# BEYOND SURFACE REASONING: UNVEILING THE TRUE LONG CHAIN-OF-THOUGHT CAPACITY OF DIFFUSION LARGE LANGUAGE MODELS

## ABSTRACT

Recently, Diffusion Large Language Models (DLLMs) have offered high through-put and effective sequential reasoning, making them a competitive alternative to autoregressive LLMs (ALLMs). However, parallel decoding, which enables simultaneous token updates, conflicts with the causal order often required for rigorous reasoning. We first identify this conflict as the core Parallel–Sequential Contradiction (PSC). Behavioral analyses in both simple and complex reasoning tasks show that DLLMs exhibit genuine parallelism only for directly decidable outputs. As task difficulty increases, they revert to autoregressive-like behavior, a limitation exacerbated by autoregressive prompting, which nearly doubles the number of decoding steps with remasking without improving quality. Moreover, PSC restricts DLLMs' self-reflection, reasoning depth, and exploratory breadth. To further characterize PSC, we introduce three scaling dimensions for DLLMs: parallel, diffusion, and sequential. Empirically, while parallel scaling yields con-sistent improvements, diffusion and sequential scaling are constrained by PSC. Based on these findings, we propose several practical mitigations, parallel-oriented prompting, diffusion early stopping, and parallel scaling, to reduce PSC-induced ineffectiveness and inefficiencies.

## 1 INTRODUCTION

In recent years, diffusion large language models (DLLMs) have emerged as a novel generative paradigm, attracting increasing research attention (Li et al., 2025; Yang et al., 2025). Represen-tative works such as LLaDA (Nie et al., 2025b) and Dream (Ye et al., 2025) adopt a two-stage mask-denoising training strategy combined with parallel decoding for masked token prediction, effec-tively mitigating the "reversal curse" in traditional autoregressive large language models (ALLMs). Mercury (Inception Labs, 2025) and Fast-DLLM (Wu et al., 2025) further demonstrate the parallel efficiency of DLLMs, achieving an impressive generation speed in code tasks.

Meanwhile, the rapid development of the Long Chain-of-Thought (Long CoT) (Guo et al., 2025; Chen et al., 2024; 2025) has spurred increasing research on applying DLLMs to extended reasoning tasks (Wang et al., 2025b;a). Zhao et al. (2025) and Tang et al. (2025) employ diffusion-augmented SFT and GRPO to further improve reasoning (Gong et al., 2025). Moreover, Trado (Wang et al., 2025b) exploits overlooked information in sampling trajectories, achieving substantial gains.

As shown in Figure 1 (a), DLLMs generate text in parallel, producing a few non-sequential words in a single diffusion step. In sequential reasoning scenarios (Figure 1 (b)), the generation of $step_i$ requires the completion of $step_{i-1}$, leading to lower entropy (Cui et al., 2025; Agarwal et al., 2025). In contrast, Figure 1 (c) shows DLLMs to parallel-decode by generating $step_{i+1}$ before $step_i$, resulting in high entropy. Nevertheless, these parallel and sequential processes are inherently contradictory: **parallelism involves simultaneous processing, while sequential reasoning requires ordered steps.** To address this, we introduce the *Parallel–Sequential Contradiction (PSC)*, which explores the underlying mechanisms and practical implications of diffusion-based reasoning.

To investigate this issue systematically, as shown in Figure 1 (d, e), we focus on two central research questions: (1) **Do DLLMs truly perform parallel reasoning that avoids PSC?** (2) **What challenges do DLLMs meet in Long CoT based on PSC?** To address the first question, we analyze

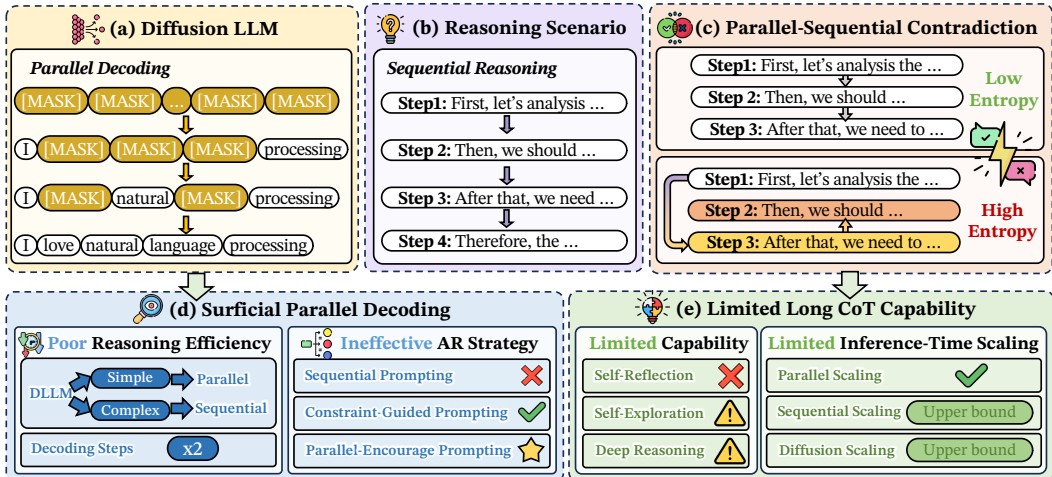

Figure 1: Overview of our work. Applying DLLMs to reasoning scenarios reveals an inherent contradiction between parallel processing and sequential reasoning, leading to high entropy, surficial parallel decoding, and limited Long CoT capabilities.

the decoding behavior of DLLMs in both simple and complex reasoning scenarios. Our findings show that DLLMs fail to achieve genuine parallel reasoning due to the PSC. They perform superficial parallel computation when outputs can be directly produced, but revert to an autoregressive mode under higher reasoning demands. This reliance on autoregression affects computational efficiency, which nearly doubles the computational cost with low confidence remasking. Furthermore, while autoregressive prompting is effective in ALLMs, it conflicts with DLLMs' parallel decoding design, amplifying the PSC of DLLMs. In contrast, strategies that reduce contradiction, such as conditional prompting or prompts that encourage parallel generation, effectively enhance prompting performance.

To understand the second question, we examine the core capabilities of Long CoT in DLLMs. Our analysis reveals that, when faced with PSC, DLLMs often demonstrate limited self-reflection, shallow reasoning depth, and constrained exploratory behavior. Furthermore, we introduce three scaling dimensions for inference time, specifically designed for DLLMs: parallel, diffusion, and sequential scaling. Our findings show that both diffusion and sequential scaling are significantly constrained by PSC, while the parallel scaling law remains unaffected due to its vertical relationship with PSC.

In summary, our key contributions are as follows:

- **Identification of Parallel-Sequential Contradictions**: To our knowledge, we first identify the Parallel-Sequential Contradiction (PSC) in DLLMs for Long CoT. We demonstrate that PSC leads to superficial parallel reasoning and reduced efficiency, requiring twice the decoding steps.
- **Systematic Exploration of DLLM Reasoning Limitation**: We conduct a systematic evaluation of DLLM reasoning, identifying the degradation of three core Long CoT capabilities, confirming the ineffectiveness of traditional autoregressive prompting methods, and demonstrating that diffusion scaling and sequential scaling are upper-bounded by PSC limitations.
- **Novel Mitigation Strategies**: We propose novel strategies to mitigate these issues and enhance DLLM reasoning. Our methods include parallel-encouraging prompting, diffusion early stopping, and parallel scaling, which substantially alleviate the constraints imposed by PSC.

## 2 PARALLEL-SEQUENTIAL CONTRADICTION

### 2.1 PARALLEL MASKED DIFFUSION LANGUAGE MODELS

In Diffusion Large Language Models (DLLMs), inference reconstructs missing spans by predicting masked tokens conditioned on a partially masked input. Its goal is modelling the conditional likelihood $p_\theta(x_0^i|x_l)$ for masked positions:

$$-\mathbb{E}_{l,x_0,x_l}\left[\frac{L}{l}\sum_{i=1}^{L}\mathbf{1}[x_l^i \in \mathcal{M}]\log p_\theta(x_0^i|x_l)\right],\tag{1}$$

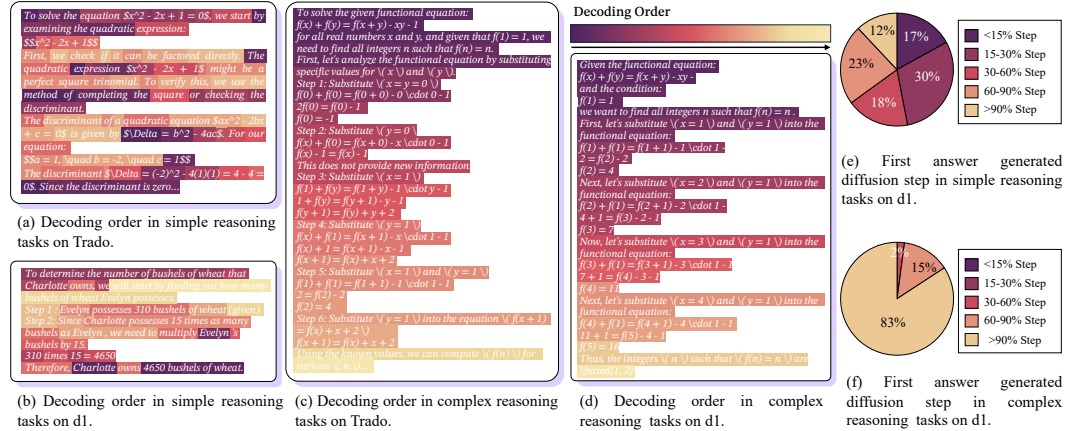

Figure 2: Diffusion order analysis with d1 (Zhao et al., 2025) and Trado (Wang et al., 2025b), where later decoding orders are indicated by shallower colors.

where $L$ denotes the total number of tokens; $l$ is the number of masked tokens, uniformly sampled from $\{1, 2, \ldots, L\}$; $x_0$ is the complete original sequence. $x_l$ is the partially masked sequence obtained by replacing those $l$ positions in $x_0$ with mask tokens, which serves as the conditional input. The indicator $\mathbf{1}[x_l^i \in \mathcal{M}]$ equals 1 if position $i$ is masked and 0 otherwise.

## 2.2 SEQUENTIAL LONG CHAIN-OF-THOUGHT REASONING

Long Chain-of-Thought (Long CoT) allows LLMs to tackle complex problems by generating a sequence of reasoning steps. This method solves a problem $P$ by following an ordered series of steps $S_1, S_2, \ldots, S_n$, leading to the final answer $A$. Formally, it can be defined as:

$$p_\theta(A|P) = \prod_{t=1}^{n+1} p_\theta(S_t|P, S_{<t}). \quad (2)$$

Here, $S_{n+1} = A$, meaning the final answer is treated as the last step of the reasoning sequence. When generating each step $S_t$, the model computes the conditional probability based on the problem $P$ and all previously generated steps $S_{<t}$.

## 2.3 PARALLEL-SEQUENTIAL CONTRADICTION

We propose the term "Parallel-Sequential Contradiction (PSC)" to capture the fundamental tension faced by diffusion models in reasoning tasks. This contradiction manifests at two levels:

- **Mechanism Level:** The inherent parallel decoding nature of diffusion models directly conflicts with the sequential dependency logic required for CoT reasoning at the computational level.

- **Behavioral Level:** This conflict causes the model's generation process to oscillate between "following low-entropy sequential logic" and "falling into high-entropy parallel guessing", reflecting an inherent inconsistency.

For **tasks with high parallelism**, downstream states typically yield predictable, high-probability outcomes, resulting in low predictive entropy. In these cases, optimizing the conditional probability $p_\theta(S_k \mid S_1)$ is efficient, making non-autoregressive or semi-parallel generation methods advantageous. In contrast, **tasks with strong sequential dependencies** exhibit high entropy when predicting distant future states in parallel. This uncertainty leads to significant predictive loss. To reduce this loss, the model is encouraged to break down the generation process into a sequence of low-entropy, step-by-step predictions. As a result, parallel generation conflicts with the model's objective of identifying a low-loss, high-probability sequential path. The formal proof is provided in Appendix B.

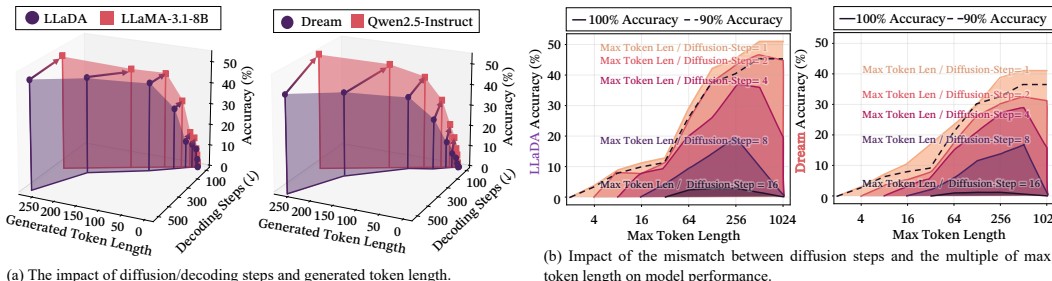

(a) The impact of diffusion/decoding steps and generated token length.

(b) Impact of the mismatch between diffusion steps and the multiple of max token length on model performance.

Figure 3: Diffusion speed analysis in Long-CoT-needed tasks with LLaDA-8B-Instruct (Nie et al., 2025a) and Dream-7B-Instruct (Ye et al., 2025) on BigGSM benchmark (Chen et al., 2024).

# 3 DO DLLMs TRULY PERFORM PARALLEL REASONING THAT AVOIDS PSC?

## 3.1 PARALLEL-SEQUENTIAL CONTRADICTIONS CAUSE SUPERFICIAL PARALLEL REASONING.

To examine whether DLLMs can genuinely perform parallel reasoning, we analyze their decoding behavior in both simple and complex scenarios. We have two key observations:

**Parallel-Sequential Contradictions cause superficial parallel reasoning in simple scenarios**  As shown in Figure 2 (a, b), DLLMs demonstrate parallel reasoning in simple cases where the model can direct output results without reasoning. For instance, when solving "$x^2 + 2x + 1 = 0$", the model may simultaneously generate the Quadratic Formula "$\Delta = b^2 - 4ac$" and the final solution "$x = -1$" within a few diffusion steps. Following this, DLLMs complete the remaining reasoning steps in parallel, demonstrating the ability to leverage diffusion-based decoding to arrive at direct solutions without relying heavily on sequential reasoning. To further explore this, we analyze the distribution of answers generated in the initial diffusion steps. As shown in Figure 2 (e), over 47% of answers are produced within the first 30% of diffusion steps. This suggests that in simple scenarios, DLLMs are capable of performing parallel reasoning, even though the underlying thought process, such as applying the Quadratic Formula before deriving the result, is inherently sequential.

**For complex reasoning, DLLMs converge toward autoregressive-like behavior to avoid PSC**  To examine DLLM behavior in complex reasoning tasks, we validate PSC where the model cannot directly output the correct answer. Figure 2 (c, d) shows that DLLMs increasingly resemble autoregressive models. For example, when addressing tasks beyond direct generation, the model defaults to an autoregressive process. This suggests difficulty in sustaining parallel reasoning, which shifts to step-by-step processing. Figure 2 (f) further confirms this observation: in complex tasks, answers emerge later in the diffusion steps, reflecting a stronger reliance on ordered reasoning. These findings indicate that DLLMs face inherent PSC challenges in balancing parallel generation with sequential reasoning, ultimately converging toward autoregressive-like processing in complex scenarios.

**PSC are relevant to stronger context dependencies, which even amplified by RL.**  Through an information-theoretic analysis of conditional entropy and a dependency index, we show that stronger contextual dependencies intensify the PSC and cause marked performance degradation in DLLMs (see Appendix J). This behavior follows from inherent limitations of the parallel generation architecture, providing additional support for our theoretical interpretation of the PSC phenomenon.Additionally, reinforcement learning-based fine-tuning further amplifies the PSC in DLLMs (see Appendix I), suggesting that common post-training methods may worsen underlying architectural mismatches. Our analysis reveals that an increase in dependency strength (i.e., lower conditional entropy or higher DI) based on RL has a significant negative impact on DLLM performance, particularly when modeling long-range dependencies is required.

## 3.2 DIFFUSION-STEP DILEMMA: SACRIFICING EFFICIENCY UNDER PSC

To investigate the reasoning efficiency of current DLLMs, we systematically categorize questions in BigGSM (Chen et al., 2024) into different sampling lengths and diffusion steps (with low-confidence remasking). We evaluate two representative DLLMs under exponentially increasing diffusion steps and max token lengths (ranging from 1 to 1024). See Appendix D for more details.

**When complex reasoning, DLLMs require significantly more diffusion steps than ALLMs.**  As shown in Figure 3 (a), achieving comparable length and accuracy to ALLMs demands over 25% more

| Model Name | BigGSM (Acc.) | GSM8K (Acc.) | Math-500 (Acc.) | HumanEval (Pass@1) | Average |
|---|---|---|---|---|---|
| Dream-7B-Instruct | $41.15_{(+0.00)}$ | $80.52_{(+0.00)}$ | $37.00_{(+0.00)}$ | $51.22_{(+0.00)}$ | $52.47_{(+0.00)}$ |
| +Zero-CoT | $41.15_{(-0.00)}$ | $77.26_{(-3.26)}$ | $34.00_{(-3.00)}$ | $48.78_{(-2.44)}$ | $50.30_{(-2.18)}$ |
| +Plan-and-Solve | $34.75_{(-6.40)}$ | $78.85_{(-1.67)}$ | $20.20_{(-16.80)}$ | $48.78_{(-2.44)}$ | $45.65_{(-6.83)}$ |
| +Least-to-Most | $35.25_{(-5.90)}$ | $77.03_{(-3.49)}$ | $16.20_{(-20.80)}$ | $43.29_{(-7.93)}$ | $42.94_{(-9.53)}$ |
| +Complex-CoT | $41.80_{(+0.65)}$ | $80.95_{(+0.43)}$ | $37.40_{(+0.40)}$ | $\mathbf{52.44}_{(+1.22)}$ | $53.15_{(+0.68)}$ |
| +MARP | $42.95_{(+1.80)}$ | $80.52_{(+0.00)}$ | $37.20_{(+0.20)}$ | $51.22_{(+0.00)}$ | $52.97_{(+0.50)}$ |
| +Diff-MARP | $\mathbf{47.21}_{(+6.06)}$ | $\mathbf{82.64}_{(+2.21)}$ | $\mathbf{43.60}_{(+6.60)}$ | $52.44_{(+1.22)}$ | $\mathbf{56.47}_{(+4.00)}$ |
| LLaDA-8B-Instruct | $48.03_{(+0.00)}$ | $75.36_{(+0.00)}$ | $34.80_{(+0.00)}$ | $32.32_{(+0.00)}$ | $47.63_{(+0.00)}$ |
| + Zero-CoT | $35.57_{(-12.46)}$ | $73.46_{(-1.90)}$ | $32.40_{(-2.40)}$ | $28.05_{(-4.27)}$ | $42.37_{(-5.26)}$ |
| + Plan-and-Solve | $31.64_{(-16.39)}$ | $72.33_{(-3.03)}$ | $29.00_{(-5.80)}$ | $27.44_{(-4.88)}$ | $40.10_{(-7.53)}$ |
| + Least-to-Most | $34.75_{(-13.28)}$ | $73.31_{(-2.05)}$ | $30.80_{(-4.00)}$ | $27.44_{(-4.88)}$ | $41.58_{(-6.05)}$ |
| + Complex-CoT | $48.03_{(+0.00)}$ | $76.50_{(+1.14)}$ | $36.20_{(+1.40)}$ | $36.59_{(+4.27)}$ | $49.33_{(+1.70)}$ |
| + MARP | $48.20_{(+0.17)}$ | $76.35_{(+0.99)}$ | $34.40_{(-0.40)}$ | $35.37_{(-3.05)}$ | $48.58_{(+0.95)}$ |
| +Diff-MARP | $\mathbf{55.74}_{(+7.71)}$ | $\mathbf{76.80}_{(+1.44)}$ | $\mathbf{38.20}_{(+3.40)}$ | $\mathbf{38.41}_{(+6.09)}$ | $\mathbf{52.29}_{(+4.66)}$ |
| LLaDA-v1.5 | $41.80_{(+0.00)}$ | $74.98_{(+0.00)}$ | $38.00_{(+0.00)}$ | $36.59_{(+0.00)}$ | $47.84_{(+0.00)}$ |
| + Zero-CoT | $36.39_{(-5.41)}$ | $71.87_{(-3.11)}$ | $37.20_{(-0.80)}$ | $35.98_{(-0.61)}$ | $45.36_{(-2.48)}$ |
| + Plan-and-Solve | $30.16_{(-11.54)}$ | $74.37_{(-0.61)}$ | $34.40_{(-3.60)}$ | $35.98_{(-0.61)}$ | $43.73_{(-4.12)}$ |
| + Least-to-Most | $35.90_{(-5.90)}$ | $73.69_{(-1.29)}$ | $34.60_{(-3.40)}$ | $31.71_{(-4.88)}$ | $43.98_{(-3.87)}$ |
| + Complex-CoT | $50.16_{(+8.41)}$ | $75.51_{(+0.53)}$ | $\mathbf{39.40}_{(+1.40)}$ | $\mathbf{39.02}_{(+2.43)}$ | $51.04_{(+3.19)}$ |
| + MARP | $42.13_{(+0.33)}$ | $74.37_{(0.61)}$ | $38.20_{(+0.20)}$ | $37.20_{(+0.61)}$ | $47.98_{(+0.13)}$ |
| +Diff-MARP | $\mathbf{54.49}_{(+12.79)}$ | $\mathbf{76.50}_{(+1.52)}$ | $42.80_{(+4.80)}$ | $38.41_{(+1.82)}$ | $\mathbf{53.08}_{(+5.23)}$ |
| LLaDOU-Math | $42.13_{(+0.00)}$ | $81.88_{(+0.00)}$ | $45.80_{(+0.00)}$ | $39.02_{(+0.00)}$ | $52.21_{(+0.00)}$ |
| +Zero-CoT | $38.52_{(-3.61)}$ | $80.95_{(-0.93)}$ | $45.80_{(-0.00)}$ | $37.80_{(-1.22)}$ | $50.77_{(-1.44)}$ |
| +Plan-and-Solve | $40.82_{(-1.31)}$ | $81.12_{(-0.76)}$ | $43.20_{(-2.60)}$ | $38.41_{(-0.61)}$ | $50.89_{(-1.32)}$ |
| +Least-to-Most | $40.16_{(-1.97)}$ | $79.08_{(-2.80)}$ | $43.00_{(-2.80)}$ | $36.59_{(-2.43)}$ | $49.71_{(-2.50)}$ |
| + Complex-CoT | $43.77_{(+1.64)}$ | $83.70_{(+1.82)}$ | $45.80_{(+0.00)}$ | $\mathbf{42.07}_{(+3.05)}$ | $52.47_{(+0.26)}$ |
| + MARP | $41.15_{(-0.98)}$ | $82.18_{(+0.30)}$ | $45.60_{(-0.20)}$ | $40.26_{(+1.24)}$ | $52.30_{(+0.09)}$ |
| +Diff-MARP | $\mathbf{54.26}_{(+12.13)}$ | $\mathbf{84.76}_{(+2.88)}$ | $\mathbf{49.00}_{(+3.20)}$ | $40.85_{(+1.83)}$ | $\mathbf{57.22}_{(+5.01)}$ |

Table 1: Performance comparison across 4 benchmarks. **Bold** marks the best baseline score per metric. For each method, we report its most token-efficient variant. Here, "⬚": prompting strategies, "⬚": offline strategies, "⬚": online strategies. See results in Table 4 with different decoding methods.

diffusion or decoding steps with remasking. In extreme cases, DLLMs require up to twice the token length in diffusion steps to match the performance and output length of autoregressive models when generating over 256 tokens. It indicates that effective reasoning entails roughly double the diffusion steps relative to the answer length, underscoring a notable efficiency challenge in reasoning tasks.

**In reasoning scenarios, a large number of diffusion steps for autoregressive reasoning is unavoidable for acceptable accuracy.** Each generated token requires a sufficient number of diffusion iterations to allow the model to reason effectively and produce high-quality outputs. As illustrated in Figure 3 (b), performance sharply declines when diffusion steps fall below the target token length. For example, generating 80 tokens with a maximum length of 128 but only 64 diffusion steps results in over a 10% accuracy drop; with 32 steps, accuracy decreases by about 40%. This demonstrates that inadequate diffusion severely impairs reasoning, as the model lacks enough refinement iterations. Thus, diffusion steps should at least match the planned token length to maintain reasoning quality. Nonetheless, excessive diffusion can significantly reduce efficiency.

## 3.3 RETHINKING THE PROMPTING STRATEGIES IN DLLMs FROM PSC PERSPECTIVE

In general, traditional autoregressive inference methods are typically categorized into two types: pipeline-guided approaches and condition-following approaches (see Appendix F for details). In this section, we will begin by reviewing the theoretical foundations and representative implementations of these two categories. We will then examine their practical limitations and challenges. Furthermore, we introduce a parallel-encouraging prompting to improve DLLM effectiveness.

**Sequential Reasoning Prompting will enlarge PSC's negative impact for DLLMs.** Sequential prompting strategies, which facilitate sequential reasoning, have been shown to significantly improve the performance of ALLMs on complex tasks. However, as indicated in purple rows of Table 1, we observed a notable decline in performance as tasks required an increasing number of reasoning steps. We attribute this decline to the fact that sequential reasoning prompts exacerbate the negative impact of PSC, thereby impairing the reasoning performance in DLLMs. Additionally, we conducted evaluations under high computational budgets. The results in Appendix F.6 indicate that even with sufficient inference resources, performance remains limited.

**Constraint-guided Reasoning Prompting enhances model performance by preventing the introduction of additional PSC.** By incorporating explicit constraints into the reasoning process, constraint-guided prompting effectively narrows the model's search space, thereby preventing the emergence of additional PSC during the reasoning process in DLLMs. This focused approach results in more accurate and reliable solutions. As shown blue rows of in Table 1, methods based on this principle, such as Complex-CoT (Fu et al., 2022) and MARP (Chen et al., 2024), demonstrate superior reasoning capabilities in DLLMs compared to traditional sequential prompting methods.

**Parallel-encouraging Prompting reduces the sequential feature so that it further improves performance.** Parallel-encouraging prompting refers to the technique of presenting multiple related tasks or questions simultaneously. This approach reduces the impact of PSC and minimizes the sequential features in the prompting process. By encouraging the model to make connections across these tasks, as illustrated in green rows of Table 1, it effectively fosters DLLMs' performance, leading to more efficient reasoning and information integration (See more examples in Appendix F.4). Leveraging the parallel processing capabilities of DLLMs, this method has the potential to significantly enhance performance, particularly in complex reasoning tasks, by promoting more comprehensive and coherent solutions (See more analysis in Appendix F.5). To inspire concrete ideas for a DLLM-friendly dataset, we discuss this in Appendix F.8 to guide future work.

> **Takeaways**
>
> 1. Due to PSC, DLLMs engage in superficial parallel reasoning and exhibit autoregressive behavior in complex scenarios, which compromises their reasoning efficiency.
>
> 2. Sequential prompts prove ineffective for DLLMs, requiring PSC-free or PSC-reduced approaches like constraint-guided and parallel-encouraging prompts to guide their operation.

## 4 What challenges do DLLMs meet in Long CoT based on PSC?

Despite impressive empirical results, DLLMs' genuine reasoning abilities and scalability under Parallel-Sequential Contradictions remain open questions. We systematically evaluate Long CoT to assess these fundamental capabilities and scaling strategies.

### 4.1 DLLMs do not have sufficient basic capabilities to support Long CoT.

Long CoT is the primary innovation in recent reasoning large language models, leveraging inference-time scaling for self-exploration, self-reflection, and deep reasoning (Chen et al., 2025). Evaluation details are in Appendix G and Table 5.

**Traditional reflection strategies are Ineffective for DLLMs.** Long CoT models always employ a self-reflection mechanism for iterative reasoning refinement. To assess its efficacy, we examine two LLM paradigms: (1) Prompting Reflection and (2) Autoregressive Forcing Reflection As shown in Figure 4 (a, b), reflection paradigms yield no significant differences from vanilla reasoning chains in semantic similarity, informativeness, or token-level entropy[1]. Though the reflection process increases entropy and reduces informativeness, it maintains over 0.95 semantic similarity to original reasoning chains. These findings suggest the reflection mechanism offers only limited surface-level optimization. Figure 4 (c) further reveals a substantial token repetition ratio compared to the original path, resulting in approximately 10% reflection-to-error responses.

**Limited Efficacy of traditional exploration strategies for novel reasoning path generation.** Exploration, a fundamental competency for complex reasoning, involves a model's ability to generate diverse and innovative solutions. To assess this potential in DLLMs, we designed experiments utilizing two strategies: (1) Prompting Exploration and (2) Autoregressive Forcing Exploration. Figure 5 (a, b) reveal that current exploration strategies offer several improvements in the novel semantic of generated reasoning processes. However, these improvements remain superficial, evidenced by a high similarity ($> 0.84$) between explored paths and original results. Furthermore, as depicted in Figure 5 (c), while the new path and explore-to-correct ratios are limited ($\sim 5\%$), they nonetheless indicate a positive, albeit constrained, effect.

---

[1]An effective reflection is generally expected to drive model toward lower entropy and higher certainty.

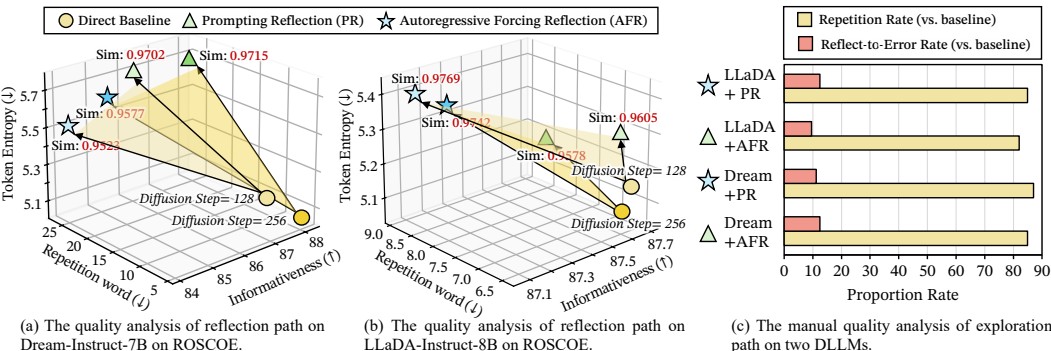

Figure 4: Self-reflection performance and rationale quality evaluation on DLLMs. In Figure (c), We conduct manual annotation for each reflection or exploration trajectory to determine whether it is repetitive compared to the preceding (baseline) trajectory (True/False) and whether its final outcome is correct (True/False), thereby obtaining the proportion of new paths or erroneous paths.

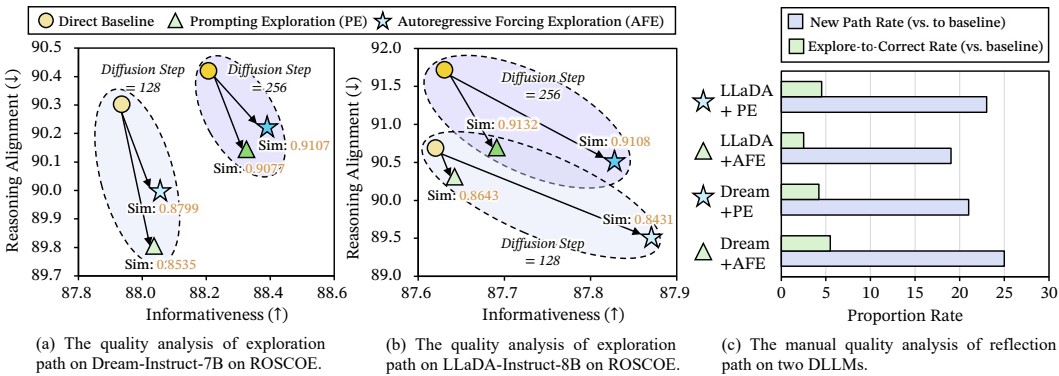

Figure 5: Self-exploration performance and rationale quality evaluation on DLLMs.

**DLLMs possess limited reasoning boundaries and, consequently, exhibit restricted deep reasoning abilities.** To examine the limitations of DLLMs on deep reasoning, we evaluate their capacity to sustain reasoning across sufficient depths. Figure 6 (a) demonstrates that error steps are all less than 2, which suggests that current DLLMs are unable to consistently sustain deep reasoning performance. Furthermore, following Chen et al. (2024), we define the 90% correctness step count as the models' completely feasible reasoning boundary (CFRB), and the 10% correctness step count as the completely infeasible reasoning boundary (CIRB). As shown in Figure 6 (b), current DLLMs display similar CFRB values but lower CIRB values, indicating narrower feasible reasoning ranges.

## 4.2 CURRENT DLLMS HAVE THREE-DIRECTIONAL BUT LIMITED INFERENCE-TIME SCALING

Given their denoising characteristics, we investigate a fundamental question: **Is there also Inference-Time Scaling Law in DLLM under such contradictions?** As shown in Figure 7, we examine this through three complementary perspectives:

- **Diffusion Scaling**: Increasing the number of denoising reasoning steps in the model. The stacked MASK blocks represent multiple, deeper iterations of denoising.
- **Parallel Scaling**: Generating multiple distinct output candidates simultaneously in parallel. The adjacent output blocks at the top represent the concurrent production of multiple output variants.
- **Sequential Scaling**: The visualization is designed to emphasize the step-by-step output generation process. The arrow direction explicitly illustrates this incremental reasoning progression.

These experiments determine whether DLLMs follow inference-time scaling laws and provide practical insights for optimizing reasoning performance. Implementation details can be seen in Appendix H.

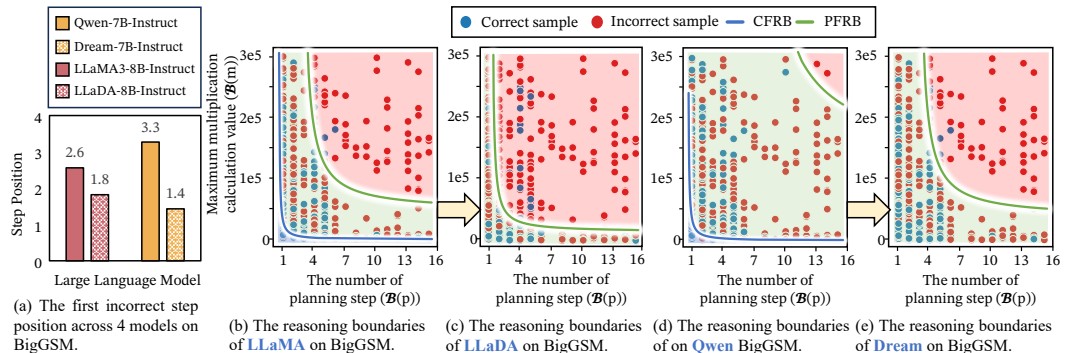

(a) The first incorrect step position across 4 models on BigGSM.

(b) The reasoning boundaries of **LLaMA** on BigGSM.

(c) The reasoning boundaries of **LLaDA** on BigGSM.

(d) The reasoning boundaries of on **Qwen** BigGSM.

(e) The reasoning boundaries of **Dream** on BigGSM.

Figure 6: Incorrect Step and Reasoning Boundaries Distribution of DLLMs on BigGSM.

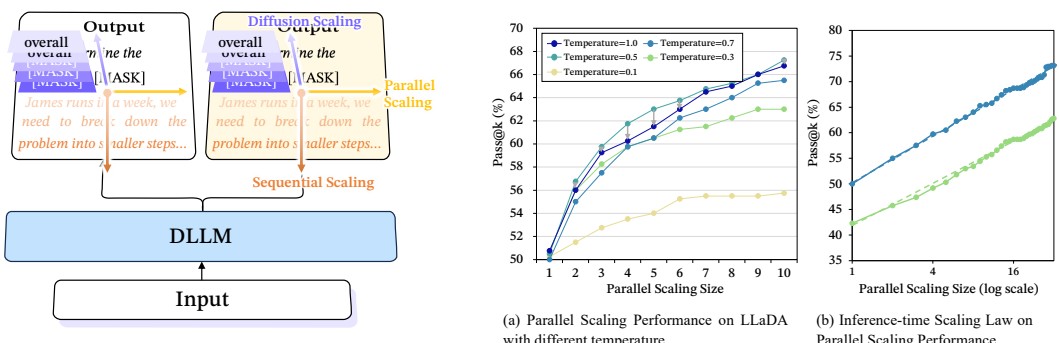

Figure 7: Three primary scaling directions for DLLMs proposed in our work.

(a) Parallel Scaling Performance on LLaDA with different temperature.

(b) Inference-time Scaling Law on Parallel Scaling Performance.

Figure 8: Performance analysis under Parallel Scaling.

#### 4.2.1 PARALLEL SCALING LAW HOLDS UNDER PSC

For DLLMs, a key question is whether their unique diffusion generation mechanism supports efficient parallel sampling and whether parallel sampling can effectively enhance reasoning performance.

**Higher temperatures do not always yield more diverse and effective parallel sampling.** The decoding temperature controls generation randomness, with higher values typically increasing output diversity in ALLM reasoning. We adjust the temperature during generation (0.1 to 1.0) to evaluate its impact on parallel sampling. Model accuracy improves steadily with increasing `Pass@k` values across all temperature settings before plateauing. As shown in Figure 8 (a), moderate temperatures (e.g., $T = 0.5$) achieve optimal performance, while both lower and higher temperatures yield diminished performance gains. This result confirms that DLLMs also obey the widely recognized trade-off governed by temperature.

**DLLM reasoning accuracy improves with increased parallel samples, following inference-time scaling patterns.** As shown in Figure 8 (b), when $k$ increases from 1 to 32, accuracy demonstrates nearly linear improvement on a logarithmic scale. This indicates that DLLMs effectively utilize parallel sampling to enhance reasoning performance, as diverse outputs increase the probability of generating correct solutions. This pattern aligns with inference-time scaling laws observed in other advanced language models, where performance scales with computational effort during inference.

#### 4.2.2 DIFFUSION SCALING IS CONSTRAINED BY PSC

**Diffusion Scaling of DLLMs ensures performance gains, with diffusion time positively correlated.** Model accuracy increases monotonically with the number of diffusion steps. We are the first to formalize *Diffusion Scaling* in DLLMs, proposing a positive correlation between model performance and diffusion iterations. To validate this claim, we benchmark two representative DLLMs, DREAM (Ye et al., 2025) and LLaDA (Nie et al., 2025a), under an exponential schedule of diffusion steps (1–1024). By tracking accuracy at each step, we observe how DLLMs address reasoning tasks of varying complexity across the diffusion process. As shown in Figure 9 (a), performance consistently

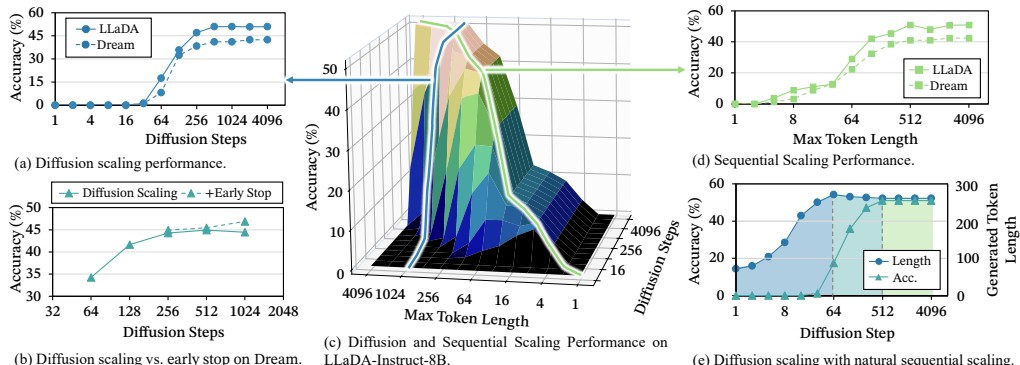

Figure 9: Diffusion Scaling analysis of reasoning accuracy across difficulty levels on BigGSM.

improves with deeper diffusion; however, the rate of improvement depends on task difficulty: simpler problems gain substantially, while tasks beyond the model's capacity yield only limited benefits.

**Diffusion Scaling is effective and exhibits an upper bound, beyond which an over-diffusion phenomenon emerges due to PSC.** Consistent with classical scaling laws, the benefits of diffusion scaling are inherently capped. As shown in Figure 9 (a), increasing diffusion steps improves performance from 32 to 512 steps, after which gains plateau. More importantly, excessive diffusion reduces accuracy: Figure 9 (b) shows a drop from 44.92% to 44.43%. This decline illustrates *over-diffusion*, where extended denoising introduces excessive corrections that disrupt reasoning chains, akin to overfitting caused by training without early stopping.

**Early stopping can effectively mitigate over-diffusion.** To address over-diffusion, we propose a Diffusion Early Stopping (DES) strategy that halts the process when generated tokens stabilize. The implementation comprises three components: (1) **Overlap Ratio Calculation**: computed as the proportion of identical tokens between consecutive steps. (2) **Convergence Detection**: potential convergence occurs when the overlap ratio meets or exceeds a predefined threshold. (3) **Activation Condition**: early stopping triggers only after three consecutive steps satisfy the threshold, preventing false positives from transient fluctuations. As shown in Figure 9 (b), we observe that beyond 256 steps, early stopping outperforms standard diffusion, with accuracy improving from 44.26% to 46.89% at 1024 steps. Early stopping captures convergence states and terminates upon stabilization, while preventing performance degradation from excessive diffusion.

### 4.2.3 SEQUENTIAL SCALING IS EQUALLY CONSTRAINED BY PSC

**The inherent limitations of sequential scaling for DLLMs.** While sequential scaling has shown promise in enhancing reasoning capabilities, it remains constrained by the inherent characteristics of DLLMs. Unlike traditional nearly unbounded diffusion scaling on pure diffusion models (Ma et al., 2025), as shown in Figure 9 (d), the performance improvements are at first increasing but eventually converge. This limitation arises from the fact that sequential scaling relies on the model's ability to maintain context over extended reasoning chains, a challenge for current DLLMs.

**Sequential Scaling also meets over-thinking challenges.** Similar to diffusion scaling, as shown in Figure 9 (d), sequential scaling faces its own set of over-thinking challenges. As the model attempts to extend its reasoning across longer contexts, it may encounter diminishing returns or even performance degradation. This phenomenon is particularly evident in tasks that require intricate reasoning over extended text, where the model's ability to track and integrate information can become strained.

**Diffusion Scaling can naturally yield Sequential Scaling benefits.** As shown in Figure 9 (e), diffusion scaling alleviates the limitations of sequential scaling. We identify three stages in DLLMs during diffusion: (1) sequential scaling, (2) compression, and (3) convergence. In the first stage, increasing diffusion steps leads to stable performance but longer solutions, indicating that DLLMs explore suitable lengths for reasoning. In the second stage, the model compresses its reasoning by eliminating redundancy, generating more efficient solutions. In the third stage, the model converges on an optimal strategy, achieving high performance while reducing computational cost.

> **Takeaways**
>
> 1. DLLMs are deficient in three basic Long CoT capabilities, hindering their effectiveness.
>
> 2. DLLM can be optimized via parallel, diffusion, and sequential scaling. Diffusion scaling inherently encompasses the benefits of sequential scaling.
>
> 3. The performance of both diffusion and sequential scaling is ultimately upper-bounded by a parallel-sequential contradiction. But Parallel scaling law remains the most effective strategy, although it is also the most computationally expensive.

## 5 RELATED WORK

The application of diffusion models to text generation has emerged as an alternative to autoregressive methods. Early work by D3PM (Austin et al., 2021) proposed discrete denoising diffusion probabilistic models, and Diffusion-BERT (He et al., 2022) demonstrated scalability to BERT-style architectures. SEDD (Lou et al., 2023) achieved performance comparable to GPT-2. Recent progress has broadened the scope of Diffusion Large Language Models (DLLMs) (Yang et al., 2025; Wu et al., 2025; Gong et al., 2025). LLaDA (Nie et al., 2025a) and Dream (Ye et al., 2025) scaled to billion-parameter models with notable inference gains. The D2F strategy (Wang et al., 2025a) further enhanced inference by enabling block-level autoregression and parallel decoding, maintaining a balance between speed and accuracy. This direction aligns with the growing interest in applying DLLMs to extended reasoning (Wang et al., 2025b; Zhao et al., 2025). Diffusion-of-Thought (DoT) (Ye et al., 2024) combines diffusion with chain-of-thought reasoning. Building on this, Zhao et al. (2025) and Tang et al. (2025) applied diffusion-augmented SFT and GRPO to strengthen reasoning. Similarly, Trado (Wang et al., 2025b) exploits overlooked sampling signals, yielding further reasoning gains.

However, while DLLMs exhibit notable parallel decoding in text generation and consistently strong step-by-step reasoning, these features appear conceptually opposed: parallelism implies simultaneous processing, whereas sequential reasoning demands ordered progression. This apparent Parallel–Sequential Contradiction (PSC) suggests that both the underlying mechanisms and the practical effectiveness of DLLMs' diffusion-based reasoning remain insufficiently understood.

## 6 CONCLUSION

In this work, we formalize the Parallel-Sequential Contradiction (PSC) to explain why DLLMs, though built for parallel decoding, revert to autoregression as reasoning difficulty rises. Empirically, DLLMs exploit parallelism only when tokens are locally decidable; otherwise, they fall back to sequential computation, reducing efficiency. Further, we first define three-dimensional scaling: parallel, diffusion, and sequential scaling, and show that PSC restricts the latter two while parallel scaling holds. We mitigate PSC through parallel-focused prompting, diffusion early stopping, and parallel scaling, improving both accuracy and throughput. Future work should align training and architectures with PSC-aware reasoning and design benchmarks, isolating its effects.

## REPRODUCIBILITY DISCUSSION

Since our paper is an analytical paper, we directly use the official DLLMs warehouse for inference. Except for the Scaling experiment, fast-dllm was added for time spent, and the others are used for inference using official inference codes, and only relevant parameters are adjusted. In addition, the relevant prompt experiments are all in Appendix F.

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

# Appendix

## A    THE USE OF LARGE LANGUAGE MODELS

In preparing this manuscript, large language models were only utilized as general-purpose writing assistants. Their role was confined to improving the clarity and refining the phrasing of the text. All scientific content, analyses, and core arguments were developed by the human authors, who take full responsibility for the final version of the paper.

## B    MATHEMATICAL PROOF OF DLLM DEGRADING TO AUTOREGRESSIVE

**Goal.**    We rigorously show, via information theory and optimization, that the intrinsic statistical property of a generative task, namely its sensitivity to perturbations of initial conditions, fundamentally determines its optimal (lowest-loss) generation strategy. Concretely, we axiomatize two classes of tasks: serial tasks (step-by-step reasoning) exhibiting cascading sensitivity to initial conditions, and parallel tasks exhibiting partial invariance, and we prove that serial tasks induce significantly higher conditional entropy for "skip-step" parallel predictions $S_k \mid S_1$ than parallel tasks, forcing any loss-minimizing learner to converge to an autoregressive strategy.

### B.1    PROBLEM SETUP AND NOTATION

Let all step states $S_i$ take values in a metric space $(\Omega, d)$.

Specifically, the true data distribution $p_\theta^S$ of a task is considered serial if, for any given $s_1 \in \Omega$, the mapping from $s_1'$ to a subsequent state $s_k'$ is highly divergent within a sufficiently small neighborhood $N(s_1, \epsilon) = \{s_1' \mid d(s_1, s_1') < \epsilon\}$. Conversely, a task's true data distribution $p_\theta^P$ is considered parallel if, for any given $s_1 \in \Omega$, there exists at least one subsequent state $S_k$ (where k>1) that is insensitive to perturbations within its neighborhood $N(s_1, \epsilon)$. Formally, this leads to the following definitions for the two tasks.

**Definition 1** (Serial tasks: cascading sensitivity with locally continuous transitions). *A data-generating distribution $p_\theta^S$ is serial if it satisfies the following properties:*

*Local continuity and learnable short-term dynamics: For each time step $t$, the conditional distribution $p_\theta^S(S_{t+1} \mid S_t = s_t)$ is **locally continuous**: there exists $\delta_t > 0$ such that for all $s_t' \in N(s_t, \delta_t)$, the distribution $p_\theta^S(S_{t+1} \mid S_t = s_t')$ has low entropy and is learnable via autoregression.*

*Sensitive long-term dependence: For any $s_1 \in \Omega$ and any $k > 1$, the compounded long-range mapping $s_1 \to s_k$ exhibits high sensitivity, satisfying:*

$$\lim_{\varepsilon \to 0} \mathbb{E}_{s_1' \sim U(N(s_1, \varepsilon))} \left[ p_\theta^S(S_k = s_k \mid S_1 = s_1') \right] = 0, \tag{3}$$

*where $s_k$ is the reference outcome drawn from the true conditional $p_\theta^S(S_k \mid S_1 = s_1)$, and $U(N(s_1, \varepsilon))$ is the uniform distribution over the $\varepsilon$-neighborhood of $s_1$.*

*This captures the dichotomy between **locally continuous and learnable short-term transitions** versus **sensitive long-term dependence** on initial conditions.*

**Definition 2** (Parallel tasks: partial invariance). *A data-generating distribution $p_\theta^P$ is parallel if there exists some $k > 1$ and a constant $C$ such that for any $s_1 \in \Omega$, the following condition satisfies:*

$$\lim_{\varepsilon \to 0} \mathbb{E}_{s_1' \sim U(N(s_1, \varepsilon))} \left[ p_\theta^P(S_k = s_k \mid S_1 = s_1') \right] = C, \tag{4}$$

*where $s_k$ is the reference outcome drawn from $p_\theta^P(S_k \mid S_1 = s_1)$, and the constant $C$ satisfies $0 < C \leq 1$. Thus, a structurally stable downstream state persists with significant probability despite infinitesimal perturbations of the initial condition.*

### B.2    LEARNING PROBLEM

Let $p_\theta$ be a parametric generative model trained by minimizing cross-entropy with respect to the true data distribution $\hat{p}$, i.e.,

$$L(p_\theta, \hat{p}) = \mathbb{E}_{x \sim \hat{p}}[-\log p_\theta(x)] = H(\hat{p}) + D_{\mathrm{KL}}(\hat{p} \| p_\theta), \tag{5}$$

so minimizing cross-entropy is equivalent to minimizing $D_{\mathrm{KL}}(\hat{p}\|p_\theta)$ and to maximum likelihood. For any conditional subproblem, the optimum satisfies $p_\theta^*(S_k \mid S_1) = \hat{p}(S_k \mid S_1)$, and the minimal expected negative log-likelihood equals the conditional entropy,

$$L^* = \mathbb{E}_{s_1 \sim \hat{p}(S_1)}\Big[H\big(\hat{p}(S_k \mid S_1 = s_1)\big)\Big], \text{ where } H(Y \mid X) \equiv -\sum_{x \in X, y \in Y} p(x,y) \log \frac{p(x,y)}{p(x)}. \quad (6)$$

**Autoregression and chain rule.** A step-by-step reaosning strategy factorizes a joint distribution as a product of conditionals via the chain rule, thereby replacing a high-entropy "skip" conditional $p_\theta(S_k \mid S_1)$ by a sequence of typically lower-entropy one-step conditionals $p_\theta(S_{t+1} \mid S_t, \ldots)$. It is the standard rationale behind likelihood-based training of ALLMs under teacher forcing.

### B.3 DISCRETIZATION

To compare entropies on a general metric space, consider a finite measurable partition $\Pi_\varepsilon$ of $\Omega$ with mesh size at most $\varepsilon$, and define the discretization and quantization map $\phi_\varepsilon : \Omega \to [m_\varepsilon]$ that assigns each $s \in \Omega$ to its cell index, where $m_\varepsilon = |\Pi_\varepsilon|$. Let $\tilde{S}_i^{(\varepsilon)} = \phi_\varepsilon(S_i)$ and write $p_\theta^{(\varepsilon)}(\cdot)$ for the induced discrete laws; we analyze $H(\tilde{S}_k^{(\varepsilon)} \mid \tilde{S}_1^{(\varepsilon)} = \tilde{s}_1)$, which is well-defined, and relate back to the original problem by taking $\varepsilon \to 0$. Two standard facts underpin the analysis: (i) for a fixed finite support, entropy is maximized by the uniform distribution; (ii) the Shannon entropy is bounded below by the min-entropy $-\log p_{\max}$, and admits tighter lower bounds in terms of the binary entropy function $H_b$ and the support size.

**Lemma 1** (Pointwise probability caps). *Fix $s_1 \in \Omega$. Under Definition 1, for any $k > 1$ and any $\eta > 0$ there exists $\delta_0 > 0$ such that for all discretization mesh sizes $\delta < \delta_0$ and for all perturbation radii $\varepsilon < \delta$,*

$$\max_{s_1' \in N(s_1, \varepsilon)} p_{\theta,S}^{(\delta)}\big(\tilde{S}_k^{(\delta)} = s_k \mid \tilde{S}_1^{(\delta)} = \phi_\delta(s_1')\big) \;\leq\; \eta, \quad (7)$$

*where $N(s_1, \varepsilon) = \{s_1' \mid d(s_1, s_1') < \varepsilon\}$ and $\phi_\delta$ is the discretization map with mesh size $\delta$.*

*Under Definition 2, there exist **some** $k > 1$, $C > 0$ and $\delta_0 > 0$ such that for all $\delta < \delta_0$ and $\varepsilon < \delta$,*

$$\max_{s_1' \in N(s_1, \varepsilon)} p_{\theta,P}^{(\delta)}\big(\tilde{S}_k^{(\delta)} = s_k \mid \tilde{S}_1^{(\delta)} = \phi_\delta(s_1')\big) \;\geq\; C. \quad (8)$$

*Proof sketch.* By Definition 1, for serial tasks, the conditional probability of a reference outcome, averaged over shrinking neighborhoods of $s_1'$, vanishes. This forces any mass that could be concentrated on a particular cell containing $s_k$ to diminish as the mesh refines. In contrast, for parallel tasks, Definition 2 guarantees a persistent mass $C \in (0, 1]$ associated with a stable outcome across neighborhoods, which uniformly lower-bounds the maximum conditional atom in $\varepsilon$.

### B.4 MAIN PROPOSITION AND QUANTITATIVE BOUNDS

**Proposition 1** (Skip-step parallel predictions on serial vs. parallel tasks). *For any $k > 1$, the optimal expected skip-prediction loss on serial-task data strictly exceeds that on parallel-task data:*

$$L_S^*\big(p_\theta(S_k \mid S_1 = s_1')\big) \;>\; L_P^*\big(p_\theta(S_k \mid S_1 = s_1')\big), \quad (9)$$

*Let $H_S$ and $H_P$ denote the conditional entropies under the Serial and Parallel data distributions, respectively. We have,*

$$\mathbb{E}_{s_1' \sim U(N(s_1, \varepsilon))}\big[H_S\big(S_k \mid S_1 = s_1'\big)\big] \;>\; \mathbb{E}_{s_1' \sim U(N(s_1, \varepsilon))}\big[H_P\big(S_k \mid S_1 = s_1'\big)\big], \quad (10)$$

*In the discrete case, this reduces to showing*

$$\mathbb{E}_{s_1' \sim N'(s_1)}\big[H_S\big(\tilde{S}_k^{(\varepsilon)} \mid \tilde{S}_1^{(\varepsilon)} = \phi_\varepsilon(s_1)\big)\big] > \mathbb{E}_{s_1' \sim N'(s_1)}\big[H_P\big(\tilde{S}_k^{(\varepsilon)} \mid \tilde{S}_1^{(\varepsilon)} = \phi_\varepsilon(s_1)\big)\big] \quad (11)$$

*with a strictly positive gap that can be quantified through discretization and classical entropy bounds.*

*Proof.* It suffices to compare the conditional entropies pointwise and then take expectations. Fix $s_1$ and a partition $\Pi_\varepsilon$. We first define the maximum of generation probability of serial tasks:

$$p_{\max}^S(\delta; s_1, k) := \max_{s_1' \in N(s_1, \varepsilon)} p_{\theta,S}^{(\delta)}\big(\tilde{S}_k^{(\delta)} = s_k \mid \tilde{S}_1^{(\delta)} = \phi_\delta(s_1')\big) \tag{12}$$

where the maximization is performed over the neighborhood of $s_1$ for a fixed target step $k$. and analogously $p_{\max}^P(\delta; s_1')$ under $p_{\theta,P}^{(\delta)}$.

By the lemma, $p_{\max}^S(\delta; s_1') \to 0$ as $\delta \to 0$, while for parallel tasks one has $p_{\max}^P(\delta; s_1') \geq C$ for all sufficiently small $\delta$. For any discrete distribution over $m_\delta$ points with maximal atom $p_{\max}$, Fano's inequality implies

$$H \leq H_b(p_{\max}) + (1 - p_{\max}) \log(m_\delta - 1), \tag{13}$$

where $H_b$ is the binary entropy.

Thus, for parallel tasks,

$$H_P\big(\tilde{S}_k^{(\delta)} \mid \tilde{S}_1^{(\delta)} = \phi_\delta(s_1')\big) \leq H_b\big(p_{\max}^P(\delta; s_1')\big) + \big(1 - p_{\max}^P(\delta; s_1')\big) \log(m_\delta - 1), \tag{14}$$

and since $p_{\max}^P(\delta; s_1') \geq C > 0$, the entropy is uniformly bounded away from the maximal value $\log(m_\delta)$ by a constant determined by $C$.

For serial tasks, since $p_{\max}^S(\varepsilon; s_1') \to 0$, we have error probability $p_e \to 0$. Now, we should apply the contrapositive of Fano's inequality. Specifically, given the Fano's inequality:

$$H \leq H_b(p_e) + p_e \log(m_\varepsilon - 1), \tag{15}$$

it follows that $H \to 0 \Rightarrow p_e \to 0$. Conversely, $p_e \to 1$ implies $H \to H_{max}$. In this sense, the condition is satisfied:

$$H_P\big(\tilde{S}_k^{(\varepsilon)} \mid \tilde{S}_1^{(\varepsilon)} = \phi_\varepsilon(s_1')\big) \to \log(m_\varepsilon - 1), \text{if } p_{\max}^S(\varepsilon; s_1') \to 0, \tag{16}$$

which reflects the extreme dispersion dictated by sensitivity. Therefore, it satisfies:

$$\mathbb{E}_{s_1' \sim N'(s_1)}\big[H_S\big(\tilde{S}_k^{(\varepsilon)} \mid \tilde{S}_1^{(\varepsilon)} = \phi_\varepsilon(s_1)\big)\big] - \mathbb{E}_{s_1' \sim N'(s_1)}\big[H_P\big(\tilde{S}_k^{(\varepsilon)} \mid \tilde{S}_1^{(\varepsilon)} = \phi_\varepsilon(s_1)\big)\big] > 0. \tag{17}$$

Q.E.D. $\square$

## B.5 CONSEQUENCES FOR OPTIMAL STRATEGY

**Formal chain rule of entropy.** To rigorously establish why sequential generation is superior for serial tasks, we invoke the chain rule of entropy:

$$H(S_k|S_1) = \sum_{t=1}^{k-1} H(S_{t+1}|S_{\leq t}). \tag{18}$$

While our proof shows that the long-range conditional entropy $H(S_k|S_1)$ is high due to cascading sensitivity, the **local continuity** assumption (revised Definition 1) ensures that each step-wise conditional entropy $H(S_{t+1}|S_{\leq t})$ remains low and learnable. The autoregressive strategy minimizes the decomposed loss:

$$L_{\mathrm{AR}} = \sum_{t=1}^{k-1} \mathbb{E}[-\log p(S_{t+1}|S_{\leq t})], \tag{19}$$

where each term has low entropy and is thus efficiently learnable. In contrast, direct parallel prediction of $p(S_k|S_1)$ must contend with the high entropy $H(S_k|S_1)$, requiring exponentially more samples to achieve the same precision.

**Optimal strategy selection.** Since the minimum expected NLL equals the conditional entropy, the high conditional entropy of skip-step predictions in serial tasks implies a high irreducible loss for any $p_\theta(S_k \mid S_1)$ objective. A loss-minimizing learner thus prefers to factorize predictions into a sequence of low-entropy one-step conditionals, i.e., an autoregressive strategy consistent with the chain-rule factorization and standard maximum-likelihood training.

By contrast, in parallel tasks, the existence of a stable high-probability outcome for some downstream state $S_k$ produces a low-entropy, high-confidence conditional, so optimizing $p_\theta(S_k \mid S_1)$ can be preferable and can support non-autoregressive or partially parallel generation plans.

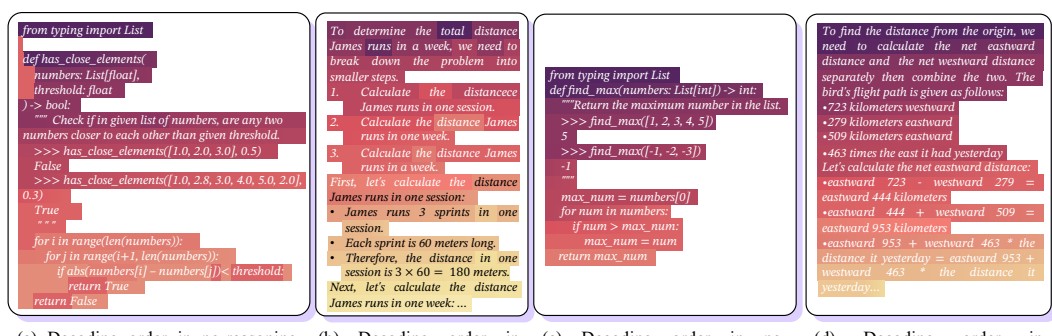

(a) Decoding order in no-reasoning tasks of LLaDA.

(b) Decoding order in reasoning tasks of LLaDA.

(c) Decoding order in no-reasoning tasks of Dream.

(d) Decoding order in reasoning tasks of LLaDA.

Figure 10: Decoding order of Dream and LLaDA on BigGSM (Chen et al., 2024).

**Takeaway.** Task-intrinsic sensitivity versus invariance dictates the shape of the optimal conditional distributions; via the cross-entropy/KL equivalence, this in turn selects the generation procedure that globally minimizes expected loss, with serial tasks forcing autoregression and parallel tasks permitting advantageous skip-step or parallel predictions.

## C  EXPERIMENT SETTING

Due to their strong fundamental performance and stable generation, this study adopts Dream-7B-Instruct (Ye et al., 2025) and LLaDA-8B-Instruct (Nie et al., 2025b) as representative diffusion language models for most experimental evaluations.

In Section 3.1, to systematically examine how PSC affects DLLM reasoning, we select D1 (Zhao et al., 2025) and TraDo (Wang et al., 2025b) for diffusion-sequence analysis: D1 refines reasoning trajectories via reinforcement learning-based reward reshaping, whereas TraDo uses dynamic programming to coordinate denoising in latent space, thereby establishing a theoretical link between noise scheduling and reasoning stability.

In Section 3.3, when assessing the effects of different prompt strategies on DLLMs, we employ LLaDA-v1.5 (Zhu et al., 2025) and LLaDOU-Math (Huang et al., 2025). LLaDA-v1.5 improves mathematical and coding abilities through dynamic mask scheduling and hierarchical denoising, while LLaDOU-Math applies a reinforcement learning-based noise scheduling scheme and attains state-of-the-art performance on the MATH benchmark.

In Section 4.1, when probing the deep reasoning capabilities of DLLMs, we include the autoregressive models LLaMA3-8B-Instruct and Qwen-7B-Instruct (Bai et al., 2023) as baselines. These two models serve as mature autoregressive references, enabling a comprehensive comparison with DLLMs.

Comprehensive experimental configurations are documented in the appendices. For Section 3.2 (Diffusion-Step Dilemma), Section 3.3 (Prompting Strategies on DLLMs), Section 4.1 (Long CoT Capability of DLLMs), and Section 4.2 (Three-directional Inference-time Scaling on DLLMs), the corresponding details are provided in Appendix D, F, G, and H, respectively.

## D  DIFFUSION-STEP EVALUATION DETAILS

In this section, we provide additional technical details on the methodology used to evaluate the impact of diffusion steps and sampling lengths in Diffusion-based Language Models (DLLMs). We specifically focus on how these parameters influence the efficiency and accuracy of the models when tackling complex reasoning tasks.

For our analysis, we utilize the BigGSM dataset (Chen et al., 2024), which includes a diverse range of complex reasoning tasks designed to test current models' ability to perform long-form reasoning. In particular, we assess the performance of two representative DLLMs on these tasks and compare them against a standard ALLM. We systematically vary both the number of diffusion steps and the sampling lengths to evaluate their combined effects on the reasoning efficiency of DLLMs. The

number of diffusion steps tested ranges from 1 to 1024, while the maximum token lengths vary from 1 to 1024 tokens, with low-confidence remasking. In each experiment, the number of diffusion steps is set equal to the maximum token length. This range allows us to assess the model's performance under different levels of token generation and diffusion refinement. For ALLMs, we adjust the maximum token length between 1 and 1024 and the temperature between 0.2 and 0.7, aiming to achieve comparable performance to that of the DLLMs.

For each setting, we track the following metrics:

- **Accuracy:** The percentage of correct answers generated by the model.
- **Model Output Length:** The number of tokens generated by the model before reaching the stopping token (calculated using the GPT-4O tokenizer).

When the maximum token length is less than or equal to 512, the model output length typically constitutes 50% to 80% of the maximum token length. Specifically, when generating a maximum token length of 512, achieving optimal performance requires 512 diffusion steps combined with low-confidence remasking strategies. We utilize this remasking approach to ensure the best performance.

Our evaluation demonstrates that the efficiency of DLLMs in reasoning tasks is strongly influenced by the balance between diffusion steps and sampling lengths. While a higher number of diffusion steps generally improves reasoning accuracy, it increases computational requirements. Thus, while sufficient diffusion steps are essential for effective reasoning, an excessive number can significantly reduce processing efficiency.

## E   THE IMPLEMENTATION OF EARLY-STOP STRATEGY

The early stopping mechanism is based on the dynamic stability of tokens, which monitors the variation of the newly updated tokens during the diffusion process to judge whether the generation has converged. We calculate the overlap ratio between the current step's selected tokens (current_tokens) and the previous step's tokens (prev_tokens) in each diffusion step. When the overlap ratio of the selected tokens remains stable over three consecutive steps and exceeds a threshold $\theta = 0.99$, early stopping is triggered. The overlap ratio is calculated as:

$$\text{overlap\_ratio} = \frac{1}{N} \sum_{j=1}^{N} \mathbb{I}(current\_tokens_j = prev\_tokens_j)$$

Where $N$ is the number of tokens updated in the current step, and $\mathbb{I}$ is the indicator function. This mechanism is controlled by the parameter early_stop_threshold = 0.99, which controls the sensitivity.The higher the threshold, the more stable the token sequence needs to be before triggering early stopping.

The parameter settings use a block-based diffusion strategy: the total generation length of 512 tokens is divided into blocks of length block_length = 32. Temperature = 0.7 helps balance exploration and exploitation. We choose low_confidence strategy, which updates tokens with low confidence. This combination ensures the quality of the generated text while improving efficiency by using fewer diffusion steps, which typically converge to a value smaller than the maximum 512 steps.

## F   PROMPTING EXPERIMENT DETAILS

### F.1   EXPERIMENTAL SETUP

In this study, we employ the following models: Dream-7B-Instruct (Ye et al., 2025), LLaDA-8B-Instruct (Nie et al., 2025b), LLaDA-v1.5 (Zhu et al., 2025), and LLaDOU-Math (Huang et al., 2025). To optimize performance, we experiment with a temperature range of [0, 1], choose top-p=0.95 and block-length=32, and select the maximum token length from the set {128, 256, 512}, as well as the diffusion step from {128, 256, 512}. For each model, we use the default decoding settings. Additionally, we apply low-confidence remasking to explore the scaling behavior. All experiments conduct on a single A100 or A800 80G GPU.

### F.2 SEQUENTIAL REASONING PROMPTING

These methods were originally designed primarily for Autoregressive Large Language Models (ALLMs) and have played a key role in optimizing their reasoning capabilities:

**Zero-CoT (Kojima et al., 2022)**   This strategy uses a simple natural-language instruction (e.g., "Let's think step by step") to elicit the inherent sequential reasoning capability of autoregressive models, enabling them to generate coherent reasoning chains without in-context examples.

> **Zero-CoT**
>
> Question: {question}
>
> Let's think step by step:

**Plan-and-Solve (Wang et al., 2023)**   By explicitly separating the reasoning process into a planning phase and an execution phase, this strategy prompts autoregressive models to first outline a solution framework and then complete the details. This improves structural integrity and global consistency, and is particularly effective for tasks requiring multi-step reasoning and long-range dependency.

> **Plan-and-Solve**
>
> Let's first understand the problem, extract relevant variables and their corresponding numerals, and make a complete plan.Then, let's carry out the plan, calculate intermediate variables (pay attention to correct numerical calculation and commonsense), solve the problem step by step, and show the answer.
>
> Question: {question}

**Least-to-Most (Zhou et al., 2022)**   In autoregressive models, this method decomposes complex problems into a sequence of simpler subproblems, guiding the model to solve them incrementally rather than tackling the full problem in a single step.

> **Least-to-Most**
>
> Q: Elsa has 5 apples. Anna has 2 more apples than Elsa. How many apples do they have together?
>
> A: Let's break down this problem:
>
> 1. How many apples does Anna have?
>
> 2. How many apples do Elsa and Anna have together?
>
> 1. Anna has 2 more apples than Elsa. So Anna has 2 + 5 = 7 apples.
>
> 2. Elsa and Anna have 5 + 7 = 12 apples together.
>
> Q: question
>
> A: Let's break down this problem:

Together, these strategies enhance the sequential reasoning behavior of autoregressive models, using prompt design to induce systematic and logically sound reasoning trajectories. However, we find that these strategies are not well suited to DLLMs.

### F.3 CONSTRAINT-GUIDED REASONING PROMPTING

**Complex-CoT:**   The original version of Complex-CoT (Fu et al., 2022) leverages a few-shot reasoning technique to prompt LLMs into performing more sophisticated reasoning processes. This approach enhances the model's ability to handle tasks that require a series of logical inferences or

multi-step reasoning, thereby improving the overall performance on complex questions. Specifically, by providing a few-shot example that demonstrates how to perform intricate reasoning, the model learns to apply similar patterns to new, unseen problems.

In contrast, the Constrained-Guided Version of Complex-CoT introduces a crucial modification to meet specific requirements. Rather than using few-shot examples, we reframe the prompting method as instruction-based, zero-shot constraints created by human experts. These constraints guide the model's reasoning process without the need for training on a set of example problems. To implement this approach, the following prompting structure is used to ensure that the model approaches each question with the necessary depth and detail:

---

**Complex-CoT (Constrained-Guided Version)**

You should think about the following question as thoroughly and in as much detail as possible.

Question: {question}

---

**MARP:** The original MARP (Chen et al., 2024) employs an instruction-based, in-context-learning approach to guide LLMs in structuring and constraining each step of the reasoning process. This method decomposes complex problems into manageable components by promoting multi-step reasoning, while ensuring each step is focused and achievable. By constraining reasoning at each stage, MARP prevents overgeneralization and ensures logical, organized outputs.

To meet the requirements of the Constrained-Guided Version, we modify MARP in two ways: first, by organizing reasoning into discrete steps, and second, by enabling parallel processing within each step. This approach allows the model to perform multiple operations simultaneously without compromising clarity or precision. The key concept is to balance step-by-step reasoning with parallel processing, enhancing task efficiency. Each reasoning step involves multiple basic operations, ensuring clarity and minimizing computational overhead.

The following prompt structure is used to guide the model's reasoning process:

---

**MARP (Constrained-Guided Version)**

Reason step by step, but process operations in parallel.

- At each step, you may perform multiple simple operations (up to 5).
- Each operation must remain basic and not involve excessive complexity.
- If you choose to perform more operations in a single step, then each operation must be correspondingly smaller in scope.

Question: {question}

---

### F.4 PARALLEL-ENCOURAGING PROMPTING

This parallel-encouraging strategy is essential for reducing PSC in reasoning. To adapt MARP for DLLM, we enable the model to process multiple operations concurrently, avoiding the bottleneck of sequential processing, where each step depends on the completion of the previous one. This parallel processing speeds up reasoning and enhances scalability. At the same time, operational complexity constraints ensure the reasoning process remains clear and manageable. This method balances parallel execution with simplicity, allowing for effective multi-step reasoning without overwhelming the model with overly complex tasks. Moreover, the model adjusts operation complexity dynamically. When tasked with more operations in a given step, each operation must be simpler, preventing cognitive overload and helping the model stay focused on individual tasks.

Ultimately, this approach enables the model to execute parallel reasoning efficiently while maintaining clarity and precision. The detailed prompting for implementation is as follows:

The key to Diff-MARP's performance improvement lies in the phrase "Reasoning in parallel," which explicitly elicits the model's parallel reasoning capability. Diff-MARP formalizes and systematizes

> **Diff-MARP**
>
> Reasoning in parallel.
> - At each step, you may carry out several small operations at the same time.
> - Keep each operation simple.
>
> Question: {question}

this instruction, constraining the number of parallel operations to avoid overload and thereby improving robustness relative to using the instruction alone. Here, we present a case using Diff-MARP.

---

**Diff-MARP Case Study: Using Diff-MARP Strategy**

Step 1: Calculate the total sleep time from 10 p.m. to 6 a.m.

Kim's usual sleep time is from 10 p.m. to 6 a.m., which is 8 hours. Since there are 60 minutes in an hour, her usual sleep time is 8 * 60 = 480 minutes.

Step 2: Calculate the time she spent sleepwalking.

Kim was sleepwalking from 2:15 to 2:38 am, which is 23 minutes.

Step 3: Calculate the time she woke up earlier than usual.

Kim woke up 5 minutes earlier than usual to go to the bathroom.

Step 4: Calculate the total time she spent on her bed that day.

The total time she spent on her bed that day is the usual sleep time minus the time she spent sleepwalking and the time she woke up earlier than usual. So, 480 minutes - 23 minutes - 5 minutes = 452 minutes.

---

**Diff-MARP Case Study: Without Using Diff-MARP Strategy**

1. Kim usually goes to sleep at 10 p.m. and wakes up at 6 a.m. This is a total of 8 hours.

2. On the day in question, she woke up 5 minutes earlier than usual to go to the bathroom.

3. She was sleepwalking from 2:15 to 2:38 am, which is a total of 23 minutes.

4. To find out how many minutes she slept on her bed that day, we need to subtract the time she was sleepwalking from the total time she was supposed to sleep.

5. Total sleep time = 8 hours * 60 minutes/hour = 480 minutes

6. Sleepwalking time = 23 minutes

7. Sleep on bed time = Total sleep time - Sleepwalking time = 480 minutes - 23 minutes = 457 minutes

8. However, she woke up 5 minutes earlier than usual, so we need to subtract those 5 minutes from the sleep on bed time.

9. Final sleep on bed time = 457 minutes - 5 minutes = 452 minutes.

---

- **Using the Diff-MARP Strategy:** We observe that each calculation step has clear labeling and explanations. This decomposition helps ensure transparency in the reasoning process, and the logical steps from start to finish are easy to follow.

- **Without Using the Diff-MARP Strategy:** When Diff-MARP is not used, although the result is the same, the steps are more condensed, and the structure of the reasoning process is weaker. The key difference lies in how Diff-MARP systematically guides the reasoning process, providing clearer explanations for each sub-step.

## F.5 Verification of PSC-Specific Enhancement

To verify that the observed performance differences arise from the PSC rather than a general limitation of DLLMs, we conducted additional experiments on the autoregressive model Qwen2.5-7B on the GSM8K dataset using the same prompts as in the main text.

| Prompting Strategy | Accuracy |
|---|---|
| Baseline | 56.8% |
| Complex-CoT | 41.5% |
| MARP | 58.8% |
| Diff-MARP | 54.2% |

Table 2: Performance of Qwen2.5-7B on GSM8K under different prompting strategies.

The results are presented in Table 2. If the performance difference stemmed only from a general capability gap, the autoregressive model should also show substantial gains under these prompts. However, its improvement is minimal or even absent. This supports our hypothesis that these strategies enhance DLLM performance mainly by mitigating PSC, rather than acting as general-purpose prompt or capability improvements.

## F.6 DLLM Performance under High-Computational Budgets

To exclude the alternative explanation that limited iteration steps hinder long-text generation, we evaluated DLLMs under high-computation budgets (max_length=1024, diffusion_steps=1024) using sequential prompts (e.g., Zero-CoT, Least-to-Most, Plan-and-Solve). As summarized in Table **??**, performance remains constrained even with substantially increased computation. This result strengthens the causal attribution of DLLM performance limits in sequential reasoning tasks primarily to PSC.

| Prompting Strategy | DREAM | LLaDA |
|---|---|---|
| Baseline | 80.6% | 75.6% |
| Zero-CoT | 75.8% | 72.3% |
| Least-to-Most | 69.3% | 73.5% |
| Plan-and-Solve | 72.4% | 71.6% |

Table 3: DLLM performance on sequential prompts under high-computation budgets.

## F.7 Eliminating the Influence of Decoding Methods

We conducted prompt experiments on the BigGSM dataset (Chen et al., 2024) using two decoding methods, entropy and topk_margin. The results shown in table 4 align with those obtained using confidence-based decoding, supporting the conclusion that PSC is an inherent property of the DLLM architecture rather than a byproduct of the default decoding method.

| Prompting Strategy | Entropy | Topk_margin |
|---|---|---|
| Baseline | 41.15 | 41.97 |
| Zero-CoT | 41.15 | 39.84 |
| Least-to-Most | 34.75 | 32.62 |
| Plan-and-Solve | 35.25 | 41.48 |
| Complex-CoT | 41.80 | 43.44 |
| MARP | 42.95 | 46.07 |
| DIFF-MARP | 47.21 | 48.19 |

Table 4: Performance comparison of different decoding methods.

## F.8 ENVISIONING IDEAL DLLM-FRIENDLY TRAINING DATA

The essence of DLLM parallel generation is global optimization and the joint generation of all tokens. Sequential reasoning requires localized and conditional generation. PSC occurs precisely between these two paradigms. Therefore, we argue that an ideal, DLLM-friendly training dataset should not follow the traditional linear chain structure but should possess the characteristic of "global conditional independence." Our envisioned design for an ideal synthetic dataset is as follows:

**Pre-planning Topology Structure**

- **Data Format:** [Question] → [Global Plan] → [Detailed Steps] → [Answer].
- **Design Principle:** The model is required to first output a high-level, low-dimensional "solution blueprint" or "global plan" in parallel before generating specific content. Once this plan is determined, it becomes the common condition for generating all subsequent detailed steps. Under this condition, the interdependency between individual steps is significantly reduced, making them suitable for parallel filling.

**Modular Parallel Structure**

- **Data Format:** Explicitly label parallel blocks, e.g., [Subtask A] ∥ [Subtask B] ∥ [Subtask C] → [Result Aggregation].
- **Design Principle:** Decompose complex problems into independently solvable submodules and explicitly declare in the data that these submodules can be processed in parallel. This forcibly guides the model to learn the independence between tasks rather than an inherent sequential order.

By systematically constructing synthetic data incorporating the above explicit independence assumptions and using it for fine-tuning the model, we can fundamentally reshape the model's attention mechanism from the root. This enables the model to better adapt to the parallel reasoning paradigm of DLLMs, thereby fundamentally mitigating the PSC problem.

# G  DLLM'S LIMITED CAPABILITIES OF LONG CoT REASONING

## G.1  LONG CHAIN-OF-THOUGHT CAPABILITIES

Following Chen et al. (2025), the Long Chain-of-Thought (Long CoT) reasoning capabilities comprise three linked components: deep reasoning, exploration, and reflection.

**Deep Reasoning.** Given $s_i$ as the $i$-th reasoning step, Deep reasoning models the conditional probability $p_\theta(s_0, s_1, \ldots, s_K | s_0)$, facilitating multi-step logical inference through iterative refinement. The associated reverse process can be characterized by a factorization:

$$p_\theta(s_0, s_1, \ldots, s_K | s_0) = \prod_{i=0}^{K} p_\theta(s_{i+1} | s_i). \tag{20}$$

**Exploration.** Exploration stems from the probabilistic nature of the reverse process. At each exploration step $s_j$, multiple samples $s_j^k$ can be drawn from the conditional distribution $p_\theta(s_j | s_i, i < j)$, enabling the model to explore diverse plausible continuations or solutions. This is formalized as:

$$s_j^k \sim p_\theta(s_j | s_i, i < j), \quad k = 1, \ldots, K, \tag{21}$$

where $K$ controls the breadth of exploration. This sampling diversity enhances robustness by covering multiple reasoning paths and mitigating premature convergence to suboptimal outputs.

**Reflection.** We view reflection as a self-correction mechanism arising from iterative conditioning on latent states. At each reverse step, the model revises its belief about the target sequence using the immediately previous state and, via the accumulated latent trajectory, all prior estimates. Formally, this corresponds to implicit message passing:

$$\hat{s}_j \sim p_\theta(s_j | s_i, i \geq j), \tag{22}$$

where $\hat{s}_j$ denotes the corrected state at step $j$, enabling iterative error correction and refinement.

Together, these components yield a procedure that combines structured logics with stochastic exploration and continual self-correction, enabling effective reasoning on complex multi-step tasks.

### G.2 STATEGIES FOR SELF-REFLECTION AND SELF-EXPLORATION EXPERIMENTS ON DLLMS

To investigate whether DLLMs truly possess the fundamental capabilities for Long CoT Reasoning, we designed two sets of experiments: self-reflection and self-exploration,using two distinct prompting strategies to examine the basic abilities of DLLMs.

**Self-Reflection**: (1)Prompting Reflection, structured reflection prompts are embedded within initial instructions,requiring the model to perform logical self-checking during generation. (2)Autoregressive Forcing Reflection, correction prompts (e.g., "Wait...there might be something wrong") are replaced with the end-of-sequence (EOS) token as a post-generation intervention strategy.

**Self-Exploration**:(1) Prompting Exploration, which embeds exploration prompts in initial instructions to activate multi-path reasoning. (2) Autoregressive Forcing Exploration, which replaces EOS token to "Let's think in another way..." to induce exploratory reasoning.

### G.3 EVALUATION OF SELF-REFLECTION AND SELF-EXPLORATION CAPABILITIES

In evaluating the self-reflection capabilities of the LLaDA-8B-Instruct (Nie et al., 2025a) and Dream-7B-Instruct (Ye et al., 2025) models, the BigGSM dataset was utilized. During the generation process, we employed a temperature of 0.7 for self-reflection and 0.2 for self-exploration, coupled with top-p sampling set to 0.95. Additionally, diffusion steps were configured to 512, and the generation length was fixed at 512. For the investigation into the models' self-exploration capabilities, the experimental settings were identical, with the sole distinction being the substitution of the reflection strategy with an exploration strategy. Based on the setting of Qin et al. (2023), we utilize the following reasoning metrics for deeper analysis:

**Semantic Alignment:** The semantic alignment metrics (Golovneva et al., 2022) lies in the *reasoning alignment vector*, which spans from the $N$-step hypothesis $\boldsymbol{h}$ to the source $\boldsymbol{s}$ of length $T$:

$$r\text{-}align(\boldsymbol{h} \rightarrow \boldsymbol{s}) = \{\alpha_1, \alpha_2, \cdots, \alpha_N\}, \tag{23}$$

where each alignment value can be calculated as:

$$\alpha_i = r\text{-}align(h_i \rightarrow \boldsymbol{s}) = \frac{\left[1 + \max_{j=1}^{T} \cos(h_i, s_j)\right]}{2} \in [0, 1]. \tag{24}$$

Here, such an alignment value is the normalized cosine similarity between the reference step and the most similar sentence in a context, and explicitly measures the *grounding* of the step-wise reasoning with respect to the source text. The alignment vector $r\text{-}align(h \rightarrow s)$ is estimated by matching the source text and the reasoning chain on the embeddings of the tokens and individual reasoning steps. A similar confidence alignment score is introduced in CTC to measure whether the information of the $i$-th source document token $s_j$ is supported by the hypothesis token $h_i$, assessing whether the reasoning step $h_i$ supports the source context $s$.

**Repetition-word:** To identify repeated, or paraphrased steps, we look at the repetition word scores (Golovneva et al., 2022) between all steps in the hypothesis chain:

$$1 - \max_{i=2\dots N} \max_{j=1\dots i-1} \left[ (1/M_i) \sum_{l=1}^{M_i} r_{\text{align}}^{\text{token}}(h_{i,l} \rightarrow h_j) \right].$$

For each pair of sentences, we look at the mean token alignment and find those sentences that maximize this alignment score. In other words, Repetition-Token will punish chains where there are at least two steps with high overlap in token embeddings.

**Informativeness:** Measures how well information present in the source is used in the reasoning steps, we calculate informativeness (Golovneva et al., 2022):

$$\frac{(1/T) \sum_{t=1}^{T} r_{\text{align}}(s_t \rightarrow h) + (1/N) \sum_{i=1}^{N} r_{\text{align}}(h_i \rightarrow s)}{2}.$$

Info-step gives a higher score to reasoning steps that are well-grounded with respect to the source, and identifies the degree of information from source that is covered by the generated hypothesis. A lower Info-Step score corresponds to the reasoning steps that are not related to the source sentences or have missed information provided in the context.

**Reasoning-Alignment** : The most straightforward way to evaluate the correctness of the hypothesis chain is to compare the degree of the overlap between the hypothesis and the reference. One way of doing that is to measure the reasoning alignment (Golovneva et al., 2022) between them:

$$\frac{1}{N} \sum_{i=1}^{N} r_{\text{align}}(h_i \to r).$$

**Token-Entropy** : To calculate the token entropy, we will utilize the pipeline as follows: First, calculate the probability of each token $p(t_i)$, which is the frequency of token $t_i$ divided by the total number of tokens $N$:

$$p(t_i) = \frac{\text{count}(t_i)}{N}$$

Next, calculate the information content $I(t_i)$ of each token, which reflects the uncertainty contribution of that token to the text:

$$I(t_i) = -\log(p(t_i))$$

Finally, token-entropy is the weighted average of the information content of all tokens, given by:

$$H = -\sum_{i=1}^{N} p(t_i) \log(p(t_i))$$

where $p(t_i)$ is the probability of token $t_i$, and $\log(p(t_i))$ is the corresponding logarithmic information content. Token-entropy reflects the overall uncertainty of the text. A higher value indicates that the text is more random and diverse, while a lower value suggests that the text is more focused and repetitive.

**Cosine similarity (Sim)** Cosine similarity measures the degree of similarity between two vectors encoded by BGE (Xiao et al., 2023) by calculating the cosine of the angle between them. For text embedding vectors, a value closer to 1 indicates greater semantic similarity. Let the two generated text vectors be $\mathbf{A}$ and $\mathbf{B}$. The formula for calculating their cosine similarity is:

$$\text{cosine\_similarity}(\mathbf{A}, \mathbf{B}) = \frac{\mathbf{A} \cdot \mathbf{B}}{\|\mathbf{A}\| \cdot \|\mathbf{B}\|} = \frac{\sum_{i=1}^{n} A_i B_i}{\sqrt{\sum_{i=1}^{n} A_i^2} \cdot \sqrt{\sum_{i=1}^{n} B_i^2}} \tag{25}$$

where $\mathbf{A} \cdot \mathbf{B}$ is the dot product of vectors $\mathbf{A}$ and $\mathbf{B}$. $\|\mathbf{A}\|$ and $\|\mathbf{B}\|$ are the Euclidean norms (magnitudes) of vectors $\mathbf{A}$ and $\mathbf{B}$. $A_i$ and $B_i$ represent the components of vectors $\mathbf{A}$ and $\mathbf{B}$ along the $i$-th dimension.

**Perplexity (PPL) of the Model** Perplexity is a concept in information theory used to measure the uncertainty of a probabilistic model in predicting samples. In natural language processing, it is employed to evaluate how well a language model fits a set of test data.

Given a sequence of $N$ tokens $W = w_1, w_2, \ldots, w_N$, where the language model predicts the probability of this sequence $P(W)$, the perplexity of the sequence is defined as:

$$\text{PPL}(W) = P(W)^{-\frac{1}{N}} = \exp\left(-\frac{1}{N} \log P(W)\right). \tag{26}$$

Because of the sequence's independence assumption, we can compute $P(W)$ as:

$$P(W) = \prod_{i=1}^{N} P(w_i | w_1, \ldots, w_{i-1}). \tag{27}$$

Therefore, the commonly seen formula for perplexity is:

$$\text{PPL}(W) = \exp\left(-\frac{1}{N} \sum_{i=1}^{N} \log P(w_i | w_1, \ldots, w_{i-1})\right), \tag{28}$$

where $\log P(W)$ is the log probability of the entire sequence. $\frac{1}{N}\sum_{i=1}^{N}\log P(w_i|w_1,\ldots,w_{i-1})$ is the average log probability of the sequence under the model. $\exp$ is the exponential function, used to transform the log probability back to its original scale.

Observation of the results for both DREAM (Ye et al., 2025) and LLaDA (Nie et al., 2025a) models, under both self-reflection and self-exploration settings, the scores across all ROSCOE-SA evaluation metrics are highly similar. This indicates that current diffusion language models (DLLMs) have not yet genuinely acquired the deeper capabilities of self-reflection and self-exploration, as their outputs do not exhibit significant differences under varying strategic prompts.

## G.4    EVALUATION OF DEEP-REASONING CAPABILITY

Following Chen et al. (2024), we further investigate reasoning boundaries (RBs) in deep reasoning capabilities in mathematical reasoning. We prompt DLLMs to generate plans and assess their accuracy through manual evaluation. When the model meets the question with fewer than 1 reasoning steps, accuracy surpasses 80%. Conversely, when reasoning steps exceed 3-4, accuracy falls below 10%. Moreover, we first randomly select 200 samples to generate examples and split steps from the DLLM-generated rationales based on ROSCOE (Golovneva et al., 2022). Further, we also manually identify the first model's incorrect step position.

## G.5    REFLECTION/EXPLORATION GAINS IN ALLMS AND DLLMS

The core inference of this section is that the limited gains from reflection and exploration in diffusion-based decision-making large language models (DLLMs) arise mainly from their inherent PSC characteristics. To test this, we conducted a comparative analysis with an autoregressive baseline under matched settings. The results in Table 5 show that the autoregressive model achieves substantially larger performance improvements, supporting the view that the weak gains of reflection and exploration are driven by the PSC issue in diffusion models rather than by intrinsic shortcomings of these methods.

|  | Qwen2.5-7B | LLaDA-8B-Instruct | Dream-7B-Instruct |
|---|---|---|---|
| Baseline | 56.8% | 52.72% | 50.7% |
| Prompting Reflection | 63.3% | 52.75% | 47.0% |
| Prompting Exploration | 63.5% | 52.75% | 48.75% |
| Autoregressive Forcing Reflection | 73.9% | 47.5% | 37.25% |
| Autoregressive Forcing Exploration | 72.1% | 52.0% | 44.5% |

Table 5: Reflection/Exploration Benefits: Autoregressive vs. Diffusion Models

# H    THREE-DIRECTIONAL INFERENCE-TIME SCALING ON DLLMS

## H.1    PARALLEL SCALING EXPERIMENT DETAILS

In the parallel scaling section, we utilize the dual-cache generation strategy from Fast-dLLM based on the diffusion language model LLaDA-8B-Instruct (Nie et al., 2025a), and perform batch processing on the BigGSM (Chen et al., 2024) dataset. Key configurations include:diffusion_steps=256, gen_length=256, block_length=32, and a Dynamic Low-Confidence Remasking mechanism.

We also employed the dual_cache generation strategy from Fast-dLLM (Wu et al., 2025) on the Dream-7B-Instruct (Ye et al., 2025) model for testing on the BigGSM reasoning dataset. The core configuration includes: diffusion_steps=256, gen_length=256, and block_length=32.

The results are shown in Figure 11. It can be observed that the accuracy generally increases with higher k-values. At the initial attempts, the accuracy at Temperature 1.0 was relatively low. Although it showed significant improvement in the early stages, its later accuracy fell behind other temperatures. At Temperature 0.1, the accuracy growth was more stable initially, but eventually plateaued at around

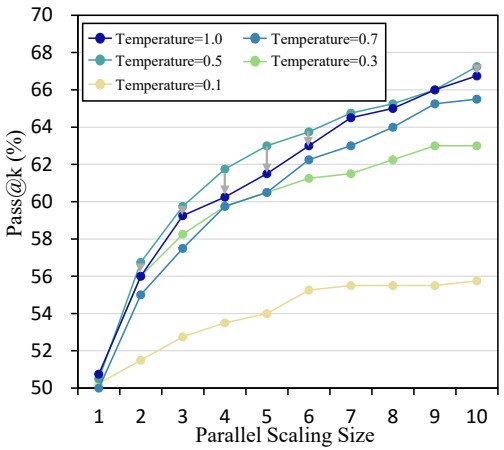 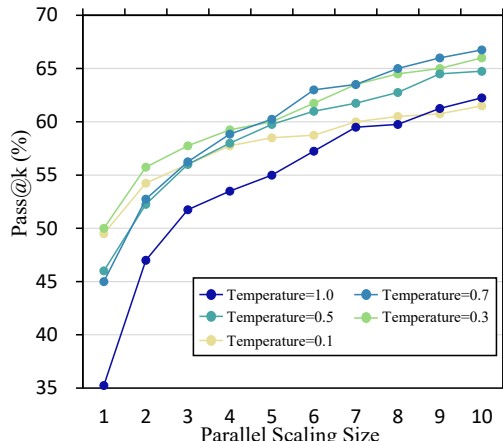

(a) Parallel Scaling Performance on LLaDA with different temperature.

(b) Parallel Scaling Performance on DREAM with different temperature.

Figure 11: Parallel scaling performance of DLLMs under Different Temperature Settings

60%, similar to Temperature 1.0. Overall, intermediate temperatures demonstrated better pass@k accuracy performance, achieving higher accuracy with more consistent and stable growth.

## H.2 DIFFUSION SCALING EXPERIMENT DETAILS

We set the diffusion_step to be between 1 and 4096 (with max-token-length equal to 512 for Figure 9 (a). The other settings are identical to those of parallel scaling. The results are shown in Figure 9 (a).

## H.3 SEQUENTIAL SCALING EXPERIMENT DETAILS

We set the Max Token Length to be between 1 and 4096 (with diffusion-step equal to max-token-length for Figure 9 (d). The other settings are identical to those of parallel scaling. The results are shown in Figure 9 (d).

## I REINFORCEMENT LEARNING INFLUENCE ON PSC

To assess the impact of post-training methods on PSC, we compare the LLaDA-8B-Instruct (Nie et al., 2025b) model with its d1 variant fine-tuned by reinforcement learning. Experiments are conducted on the BigGSM dataset (Chen et al., 2024), from which 100 samples are randomly selected. Under identical inference hyperparameters, we examine the decoding order during generation. We quantify PSC using the following two metrics:

**Inversion Pair Ratio:** During generation, if a token at an earlier position is produced after a token at a later position (i.e., if $i < j$, but the generation step $\text{step}[i] > \text{step}[j]$), it is counted as an inversion pair. A higher ratio indicates more severe sequential conflict in decoding.

**Spearman's Rank Correlation Coefficient:** This metric measures the consistency between the model's generation order and the natural left-to-right language order. A higher coefficient indicates that the generation process more closely follows habitual language sequencing.

| Model | Inversion Pair Ratio | Spearman Coefficient |
|---|---|---|
| LLaDA-8B-Instruct | 3.41% | 98.62% |
| d1 (After RL Training) | **6.65%** $_{(+3.24)}$ | **97.27%** $_{(-1.35)}$ |

Table 6: Comparison of PSC Phenomenon Before and After RL Training

The results show that, after reinforcement learning training, the model's inversion pair ratio increases from 3.41% to 6.65%, while the Spearman coefficient decreases from 98.62% to 97.27%. This suggests that reinforcement learning intensifies sequential conflicts during decoding and makes the PSC phenomenon more pronounced.

Here, we explain why the inversion pair ratios remain relatively low and the Spearman coefficients remain close to 1. Diffusion language models are designed to generate coherent text sequences that follow linguistic conventions, rather than arbitrary permutations of words. To this end, the model architecture (for example, the Block mechanism) guides generation at a coarse level to follow the sequential structure of natural language. In particular, the generation order between different blocks is strictly constrained to proceed from left to right. Consequently, both the inversion ratio and the Spearman coefficient still reflect strong overall adherence to sequential generation.

## J  THE RELATIONSHIP BETWEEN CONTEXT DEPENDENCY AND PSC

Establishing how context dependency strength relates to DLLM performance is key to characterizing the PSC phenomenon. We quantify this relationship via an information-theoretic analysis, using conditional entropy to measure context dependency. Specifically, for each sample in the BigGSM dataset, we compute the conditional entropy at each generation step with a GPT-2 model and then average these values across steps to obtain the sample's average conditional entropy.

The conditional entropy for token $x_i$ given context $x_{<i}$ is:

$$H(x_i|x_{<i}) = -\sum_{w \in V} P(w|x_{<i}) \cdot \log P(w|x_{<i}). \tag{29}$$

The average conditional entropy over a text of length $T$ tokens is:

$$\text{Avg}_{\text{Entropy}} = \frac{1}{T-1} \sum_{i=1}^{T-1} H(x_i|x_{<i}), \tag{30}$$

which quantifies how strongly the generated answer is constrained by the context: lower entropy indicates stronger contextual constraints and thus higher dependency.

Using this metric, we partition the BigGSM dataset into three subsets and compute, for each subset, the average conditional entropy and the Dream accuracy, which reflects DLLM performance.

| Entropy Level | Sample Size | Avg. Entropy | Dream Acc | Avg. DI |
|---|---|---|---|---|
| Low Entropy | 203 | 2.5269 | 13.79% | 27.41 |
| Medium Entropy | 203 | 3.0766 | 35.29% | 21.56 |
| High Entropy | 203 | 3.4951 | 45.32% | 19.06 |

Table 7: Relationship between Context Dependency Strength and DLLM Performance

The results are summarized in Table 7. As contextual dependency strengthens (i.e., conditional entropy decreases), DLLM performance drops sharply from 45.32% to 13.79%. In addition to conditional entropy, we introduce the Attention Distance Index (ADI) to further examine how dependency strength affects DLLM performance. The ADI measures whether the model's attention at each layer preferentially targets long-distance tokens, thus characterizing its ability to model strong dependencies.

We define the **Attention Distance Index (ADI)** to quantify the model's tendency to attend to long-distance tokens. For each layer, the ADI is calculated as:

$$\text{ADI}_{\text{layer}} = \frac{1}{H} \sum_{h=1}^{H} \left[ \sum_{i=1}^{L} \sum_{j=1}^{L} A_{h,i,j} \cdot \frac{|i-j|}{L-1} \right], \tag{31}$$

where $A_{h,i,j}$ is the attention weight from query token $i$ to key token $j$ in head $h$; $L$ is the sequence length; $H$ is the number of attention heads; and $\frac{|i-j|}{L-1}$ is the normalized distance between tokens.

**Dependency Index (DI)** for the entire model is then computed as the average ADI across all layers:

$$\text{DI} = \frac{1}{N} \sum_{l=1}^{N} \text{ADI}_l, \tag{32}$$

where $N$ is the total number of layers. A higher DI indicates stronger long-range dependency modeling requirements.

Our analysis shows that stronger dependencies (lower conditional entropy or higher DI) markedly degrade DLLM performance, particularly when long-range dependencies are required. This behavior is consistent with intrinsic limitations of the parallel generation architecture and further supports our theoretical interpretation of the PSC phenomenon.

