# OpenReview forum: "Beyond Surface Reasoning: Unveiling the True Long Chain-of-Thought Capacity of Diffusion Large Language Models"
_ICLR.cc/2026/Conference — ICLR 2026 Conference Desk Rejected Submission_

### Official Review · Reviewer_eVwU · 2025-10-29

**Soundness:** 3
**Presentation:** 2
**Contribution:** 2
**Rating:** 4
**Confidence:** 3

**Summary:**

The paper investigates the reasoning ability of Diffusion Large Language Models (DLLMs) and introduces a key limitation in DLLMs called the Parallel–Sequential Contradiction (PSC).
Although diffusion decoding allows parallel generation that is not constrained by token order, the authors find that in complex reasoning tasks, DLLMs gradually degenerate into sequential decoding, requiring many more steps to reach the same quality as autoregressive LLMs.

They perform extensive experiments comparing DLLMs on simple and complex tasks and discover that, 1) DLLMs only show true parallelism on trivial or directly decidable tasks; 2) for long reasoning chains, DLLMs behave like autoregressive models; 3) typical Chain-of-Thought, reflection, and exploration strategies effective for LLMs fail to help DLLMs.

They also analyze scaling laws and find that the Parallel Scaling Law still holds; the Diffusion and Sequential Scaling Laws break under PSC constraints.

Finally, the authors show that constraint-guided prompting and parallel-encouraging prompting can partially mitigate PSC, improving reasoning accuracy and efficiency.

**Strengths:**

- Very comprehensive experiments across multiple benchmarks and models (Dream-7B, LLaDA-8B, Qwen, etc.).
- Clear identification of the Parallel–Sequential Contradiction that explains why DLLMs fail on long reasoning chains.
- Empirical results are consistent and carefully analyzed.
- The paper provides a valuable empirical foundation for future work on diffusion-based reasoning models.

**Weaknesses:**

The paper has two main weaknesses.

**(1) Limited methodological contribution.**
The main strength of this paper lies in its rigorous and systematic analysis of DLLMs’ weaknesses.
The findings are convincing and the empirical evidence is solid.
However, the work remains largely *diagnostic*: it identifies the problem of the Parallel–Sequential Contradiction (PSC) but does not propose concrete or innovative solutions.
From my perspective, the paper’s value is primarily in defining and characterizing the issue, while the exploration of how to mitigate or resolve PSC is minimal.
For an ICLR-level contribution, the lack of a novel method or substantial algorithmic proposal weakens the overall impact.

**(2) Insufficient clarity and missing experimental details.**
The figures and wording of this paper are good, but many critical experimental details are missing or scattered across sections, making it hard to fully understand the setup and reproduce the results.
In particular:
- The paper does not provide a unified description of the experimental configuration.
  For example, Section 3.3 introduces three prompting strategies but doesn't specify how these strategies are implemented.
- The models used (e.g., Dream-7B-Instruct, LLaDA-8B-Instruct, LLaMA-3.1-8B) are not introduced in a setup section; they just appear inside result tables.
- In Section 4.1, the reflection and exploration strategies are mentioned but not explained — it is unclear how they are applied to baseline DLLMs.
- Figure 6(a) refers to the “error step” metric, but its definition is not provided.

Overall, the paper’s experimental insights are strong, but the missing methodological and experimental details significantly reduce readability and reproducibility.

**Questions:**

I believe this paper has meaningful value and could become a strong contribution with further clarification and expansion.
To improve the clarity and completeness of the work, I suggest the authors address the following points:

1. Provide a unified and detailed experimental setup. (see the weakness part for the details)

2. It could be a major contribution if the author could expand Section 3.3 on the mitigation strategies.

---

> ### Author Response · Authors · 2025-11-24
> **Resonse to Reviewer eVwU**
>
> We sincerely thank you for your valuable comments and recognition of our work. As per your suggestions, we have responded to each point below.
>
> > **Question 1: ​**The paper does not provide a unified description of the experimental configuration.
>
> **Answer 1: Thanks for your constructive feedback. ​**In the revised version of the paper, we have added Appendix C as the experimental setup section, clearly indicating the corresponding configuration appendix for each part of the experiments and citing it in the main text.
>
> > **Question 2: ​**Section 3.3 introduces three prompting strategies but doesn't specify how these strategies are implemented.
>
> **Answer 2:** Thank you for your valuable feedback. We sincerely believe that there are some misunderstandings. Regarding the three prompting strategies, Constraint-guided Reasoning Prompting (Complex-CoT, MARP) and Parallel-encouraging Prompting (Diff-MARP) have already provided with detailed examples in Appendix F.3. Moreover, to more completely offer implementation details, we adopted the official prompts from the original papers for Sequential Reasoning Prompting (Least-to-Most, Zero-CoT, Plan-and-Solve), which are now presented in Appendix F.2 of the revised manuscript.
>
> > **Question 3:** The models used (e.g., Dream-7B-Instruct, LLaDA-8B-Instruct, LLaMA-3.1-8B) are not introduced in a setup section; they just appear inside result tables.
>
> **Answer 3: ​**Thank you for pointing out this important omission. We have  added a dedicated section for experimental settings (Appendix C) to introduce the models used in our evaluation. This section now clearly lists and briefly describes all models to enhance the clarity and reproducibility of our work.
>
> > **Question 4: ​**In Section 4.1, the reflection and exploration strategies are mentioned but not explained — it is unclear how they are applied to baseline DLLMs.
>
> **Answer 4: ​**Thank you for insightful comment. For clearer setups, we have mentioned the implementation of reflection and exploration strategies in Appendix G.2. We categorized them into two approaches: prompting and forcing.
>
> * Prompting  Reflection/Exploration involves embedding reflective or exploratory cues into the initial instructions to guide the model's reasoning.
> * Autoregressive Forcing Reflection/Exploration refers to replacing the end-of-sequence token with guiding phrases (e.g., ​"Wait...there might be something wrong"​or ​"Let’s think in another way..."​) after the model generates its initial response, thereby forcing a new round of generation.
>
> > **Question 5: ​**Figure 6(a) refers to the “error step” metric, but its definition is not provided.
>
> **Answer 5: ​**Thank you for detailed feedback.**​ ​**The "incorrect step" mentioned here refers to the first step position where an error occurs within the reasoning trace generated by the DLLM, which is identified by manually examining the steps decomposed according to the ROSCOE framework. We have detailed this process in Appendix G.4.
>
> > **Question 6: ​**Limited methodological contribution.
>
> **Answer 6: ​**Thank you for your kind comment.**​ ​**We fully understand your comment regarding the limited methodological contribution. However, we sincerely would like to clarify that this paper is ​**positioned as an analytical study**​, with its core objective being a **discovery, ​**​​**in-depth diagnosis**​**​,​**​**​ ​**​**proof ​**​**and analysis of the PSC** to provide the research community with thoughtful insights.
>
> As a natural extension of the analytical work, we have also explored ​**preliminary mitigation directions**​. In addition to the **Parallel-encouraging Prompting strategy** proposed in Section 3.3, we first investigated the**​ three-dimensional**​**​ parallel scaling approach** in Section 4.2.1 and validated the effectiveness of the**​ early stopping strategy** in Section 4.2.2. These exploratory approaches aim to provide actionable starting points for future research.
>
> Hope our clarifications can address your concerns, and we **have followed your suggestions** to enrich our work point-by-point in the revised version. **We sincerely hope that you can reconsider our work in light of these clarifications**.

---

### Official Review · Reviewer_iUzq · 2025-10-30

**Soundness:** 2
**Presentation:** 3
**Contribution:** 2
**Rating:** 4
**Confidence:** 1

**Summary:**

This paper investigates the performance of DLLMs in Long CoT reasoning, identifying the PSC as a core limitation to their reasoning capabilities. The study reveals that while DLLMs exhibit superficial parallel reasoning in simple tasks, they revert to autoregressive-like behavior in complex scenarios. Based on these findings, the authors propose three inference-time scaling directions and demonstrate that PSC significantly constrains the latter two. Finally, the paper introduces several mitigation strategies.

**Strengths:**

The paper systematically defines and analyzes the PSC problem, uncovering a critical bottleneck for DLLMs in complex reasoning tasks.

Mathematical derivations convincingly demonstrate how PSC causes DLLMs to degrade into autoregressive behavior in sequential tasks

**Weaknesses:**

Lack of Ablation Studies: While multiple strategies are proposed, the paper lacks sufficient ablation experiments to quantify the individual contributions of each approach

Redundant Analysis and Conclusions: Some parts of the experimental analysis and conclusions appear superfluous or lack meaningful insight. For instance, the statement, "This indicates that moderate temperature settings provide the optimal balance between generation diversity and output reliability".

**Questions:**

Why is the statement “This indicates that moderate temperature settings provide the optimal balance between generation diversity and output reliability” mentioned? How does it differ from what is commonly known in the field?

The use of $\in (0,1]$ in Equation 4 of the Appendix appears redundant and unconventional. However, this does not significantly impact the overall evaluation of the paper.

I roughly understand what this paper is doing, but I am not sure if I fully grasp its practical value. My confidence in evaluating this aspect is low.

---

> ### Author Response · Authors · 2025-11-24
> **Response to Reviewer iUzq**
>
> We sincerely thank you for your valuable comments and recognition of our work. As per your suggestions, we have responded to each point below.
>
> > **Question 1:** Why is the statement “This indicates that moderate temperature settings provide the optimal balance between generation diversity and output reliability” mentioned? How does it differ from what is commonly known in the field?
>
> **Answer 1: ​**Thank you for your thoughtful comment. We agree that this is a commonly recognized concept. We mentioned it in the paper to illustrate that DLLMs also adhere to this principle, confirming that certain fundamental rules remain applicable even under vastly different generation mechanisms due to its irrelevance to PSC. This serves as part of our systematic analysis of DLLMs. In the revised version, we will rephrase the text to explicitly emphasize that we "validated this balance in DLLMs" rather than proposing the principle itself, so as to avoid any misunderstanding.
>
> > **Question 2: ​**The use of Formula Four in the appendix, which is ∈(0,1], appears redundant and unconventional.
>
> **Answer 2: ​**Thank you for your valuable suggestion. We have revised the content accordingly by moving the notation "C ∈ (0,1]" from the equation to the text description below, ensuring the equation remains concise and adheres to academic conventions while maintaining rigor. Specific details can be found in Appendix B.
>
> > **Question 3: ​**While multiple strategies are proposed, the paper lacks sufficient ablation experiments to quantify the individual contributions of each approach.
>
> **Answer 3: ​**We thank the reviewer for their valuable feedback. However, we believe there is a misunderstanding. The three methods proposed in this paper (parallel-oriented prompting, diffusion early stopping, and parallel scaling) are independent solutions for DLLMs, rather than components of a single integrated framework. All relevant experiments are designed as ablation studies to fairly quantify the individual contribution of each approach.
>
> Thus, the core contribution of this work lies in exploring and validating novel approaches to the PSC problem from different perspectives, with the goal of demonstrating the standalone effectiveness of each technical pathway. We fully agree that investigating the synergistic effects of these methods is highly valuable, and we will add more discussion in future work section.
>
> > **Question 4: ​**I roughly understand what this paper is doing, but I am not sure if I fully grasp its practical value.
>
> **Answer 4: ​**Thank you for the reviewer's question. The core value of this paper lies in **being the first to systematically reveal and define the fundamental bottleneck in diffusion large language models for sequential reasoning**, **the Parallel-Sequential Contradiction (PSC**), and to propose practical solutions. Specifically, we identify an inherent conflict between the model's parallel generation mechanism and the sequential demands of reasoning. This conflict not only leads to "pseudo-parallelism" in complex tasks, nearly doubling decoding steps and reducing efficiency, but also limits core capabilities such as self-reflection, reasoning depth, and exploratory breadth. To address this, we propose targeted strategies including**​ parallel-oriented prompting, diffusion early stopping, and parallel scaling**, which are experimentally proven to effectively mitigate this contradiction. This work **provides a theoretical foundation for understanding the reasoning limitations of diffusion models and offers concrete technical pathways to enhance their practical utility.**
>
> Hope our clarifications can address your concerns, and we **have followed your suggestions** to enrich our work point-by-point in the revised version. **We sincerely hope that you can reconsider our work in light of these clarifications**.

---

### Official Review · Reviewer_U4VP · 2025-11-01

**Soundness:** 3
**Presentation:** 3
**Contribution:** 3
**Rating:** 6
**Confidence:** 3

**Summary:**

The paper argues that DLLMs face an inherent Parallel–Sequential Contradiction (PSC): their parallel token updates clash with the ordered dependencies required by long CoT reasoning. Empirically, DLLMs appear parallel only on simple/locally decidable cases, but revert to autoregressive-like behavior on harder problems. The authors further study “inference-time scaling” along three axes, claiming only parallel scaling reliably improves accuracy. Also, they propose three mitigations: parallel-oriented prompting, diffusion early stopping, and parallel scaling.

**Strengths:**

The experiments are complete, effective, and well-executed, covering multiple prompt families, reasoning benchmarks, and DLLM variants.

The insights conveyed by the paper also appear internally logical and plausible from the narrative built by the authors.
However, since I am not an expert on diffusion-based LLMs, I cannot fully judge whether these insights are already well-known to the community or whether they represent genuinely novel conceptual ground.

**Weaknesses:**

**W1 (Attribution of reflection/exploration failures).** The paper attributes weak gains from reflection/exploration to PSC, but similar limitations often hold for autoregressive LLMs as well. Therefore these results are not specific evidence for PSC. A direct AR baseline demonstrating materially larger reflection/exploration gains (under matched settings) is needed to isolate a diffusion-specific effect.

**W2 (Cross-paradigm baselines are thin).** Many claims hinge on DLLM-specific pathology, yet the comparisons against strong AR models (with the same prompts, compute budgets, and answer-length controls) are not comprehensive enough to rule out a simpler story—namely, capability gap rather than PSC. For example, sequential prompts hurting DLLMs could be confounded by longer outputs vs. limited diffusion iterations rather than contradiction per se. Stronger AR-vs-DL baseline ablations would sharpen the causal reading.

**Questions:**

**Q1. Dependency strength quantification.**
Language always has contextual dependence, and DLLMs already show the ability to model such dependence. Reasoning tasks simply impose stronger step-to-step dependence. The paper claims such stronger dependence fundamentally limits DLLMs. Can you provide a more rigorous theoretical / empirical link between dependency intensity and performance? For example, define a quantitative dependency index (conditional entropy / mutual information / perturbation-based sensitivity), and quantify the severity of DLLM PSC manifestations as a function of this dependency index (i.e., plotting PSC strength vs. dependency intensity).

**Q2. Data-side mitigation.**
Beyond prompting and decoding, could PSC be mitigated via dataset design? If synthetic datasets can be constructed, what would the ideal DLLM-friendly training data look like?

Note: The two questions I raised above are **not additional weaknesses**.

---

> ### Author Response · Authors · 2025-11-24
> **Response to Reviewer U4VP (Part1/3)**
>
> We sincerely thank you for your valuable comments and recognition of our work. As per your suggestions, we have responded to each point below.
>
> > **Question 1:** The paper attributes weak gains from reflection/exploration to PSC, but similar limitations often hold for autoregressive LLMs as well.
>
> **Answer 1:** Thank you for your insightful comments. We fully agree with the perspective that an autoregressive baseline is necessary to isolate the unique effects of the diffusion model. Our experimental results demonstrate that, under matched settings, the ​**performance gain from reflection/exploration is indeed significantly greater for the autoregressive model than for the diffusion model**​. This finding directly validates the reviewer's key point and strengthens the paper's inference that PSC is the primary limiting factor.
>
> | GSM8K    | Qwen2.5-7B | LLaDA-8B-Instruct | Dream-7B-Instruct |
> | ---------- | ------------ | ------------------- | ------------------- |
> | baseline | 56.8%      | 52.72%            | 50.7%             |
> | reflect1 | 63.3%      | 52.75%            | 47%               |
> | explore1 | 63.5%      | 52.75%            | 48.75%            |
> | reflect2 | 73.9%      | 47.5%             | 37.25%            |
> | explore2 | 72.1%      | 52%               | 44.5%             |
> ||
>
> Table 1: Reflection/Exploration Benefits: Autoregressive vs. Diffusion Models
>
> Note: The prompts used in the additional experiments are consistent with those in the paper.
>
> * reflect/explore1: Embedding reflective/exploratory cues within the prompt.
> * reflect2: After generating the initial response, inserting "Let's think in another way..."
> * explore2: After generating the initial response, inserting "There may be something wrong..."
>
> We have supplemented this set of experiments in Appendix G.5.
>
> > **Question 2:** Many claims hinge on DLLM-specific pathology, yet the comparisons against strong AR models.
>
> **Answer 2: ​**Thanks for your thoughtful feedback. In our paper, we proposed three prompting strategies tailored for DLLMs: Complex-CoT, MARP, and Diff-MARP. To verify that the observed performance differences indeed stem from the PSC rather than general model capability gaps, we conducted additional experiments on the GSM8K dataset using Qwen2.5-7B with the same prompts as in the main text.
>
>
> |           | Qwen2.5-7B |
> | ------------- | ------------ |
> | baseline    | 56.8%      |
> | complex-CoT | 41.5%      |
> | MARP        | 58.8%      |
> | DIFF-MARP   | 54.2%      |
> ||
>
> Table 2:Performance of Qwen2.5-7B on GSM8K under different prompting strategies.
>
> Combined with the experimental data on DLLMs presented in the paper, it is evident that **the performance gains from these three prompting methods are significantly greater for DLLMs than for autoregressive ​**​​**LLMs**​. If the performance difference were solely due to model capability gaps, the autoregressive model should also exhibit substantial improvement under MARP prompting, yet the actual gains are limited. This supports our inference that these strategies enhance DLLM performance specifically by mitigating the PSC, rather than through general prompt optimization or capability enhancement.
>
> We have supplemented this experiment in Appendix F.5
>
> Furthermore, even under high computational budgets (generation length = 1024, iteration steps = 1024), DLLMs show limited improvement with sequential prompting strategies (such as Zero-CoT, Least-to-Most, and Plan-and-Solve, which are widely verified on ALLMs), as demonstrated below:
>
>
> |              | Dream-7B-Instruct | LLaDA-8B-Instruct |
> | ---------------- | ------------------- | ------------------- |
> | baseline       | 80.6%             | 75.6%             |
> | Zero-CoT       | 75.8%             | 72.3%             |
> | Least to most  | 69.3%             | 73.5%             |
> | Plan and solve | 72.4%             | 71.6%             |
> ||
>
> Table 3:DLLM performance on sequential prompts under high-computation budgets.
>
> This result rules out the alternative explanation that "limited iteration steps cause difficulties in long-text generation," further strengthening the causal inference that PSC is the performance bottleneck. We have supplemented this experiment in Appendix F.6.

---

> > ### Author Response · Authors · 2025-11-24
> > **Response to Reviewer U4VP (Part2/3)**
> >
> > > **Question 3:** Can you provide a more rigorous theoretical / empirical link between dependency intensity and performance?
> >
> > **Answer 3:** Thank you for raising this insightful question. We also sincerely believe that establishing a quantitative relationship between contextual dependency strength and DLLM performance is crucial for understanding the nature of the PSC phenomenon.
> >
> > We validate this relationship through a quantitative analysis based on information theory. First, we use conditional entropy as a metric to quantify contextual dependency strength. Specifically, for each sample in the BigGSM dataset, we employ the GPT-2 model to calculate the conditional entropy at each generation position, then take the average across all positions to obtain the average conditional entropy for that sample. This metric reflects the degree to which the generated answer depends on the context: **a lower entropy value indicates stronger**​ sequential**​ constraints provided by the context, implying stronger dependency.**
> >
> > Based on the above metric, we divided the BigGSM dataset into three subsets (each containing 203 samples) and calculated the average conditional entropy and Dream accuracy (DLLM performance) for each subset. The results are shown in the table below.
> >
> >
> > |             | Number of Samples | Average Entropy | Dream acc | Average DI |
> > | ---------------- | ------------------- | ----------------- | ----------- | ------------ |
> > | Low Entropy    | 203               | 2.5269          | 13.79%    | 27.41      |
> > | Medium Entropy | 203               | 3.0766          | 35.29%    | 21.56      |
> > | High Entropy   | 203               | 3.4951          | 45.32%    | 19.06      |
> > ||
> >
> > Table 4:The relationship between context dependency and PSC
> >
> > **We observed that as contextual dependency strength increases (i.e., conditional entropy decreases), DLLM performance declines significantly (from 45.32% to 13.79%).**
> >
> > In addition to conditional entropy, we also introduced the Attention Distance Index (ADI) to further analyze the impact of dependency strength on DLLM performance. The ADI quantifies whether the model's attention at each layer tends to focus on long-distance tokens, thereby reflecting the model's ability to handle strong dependencies.
> >
> > * **ADI Calculation Method:** For the attention matrix of each layer, we compute the attention weights of each token to all other tokens. For each attention weight, we calculate the product of the weight and the corresponding distance (i.e., the relative positional difference between tokens), then sum these products. This sum represents the ADI for that layer.
> > * **DI Calculation Method: ​**For each sample, we calculate the ADI for all layers and then average these ADI values to obtain the sample's DI. A higher DI indicates a greater need for modeling long-range dependencies.
> >
> > Our analysis reveals that increased dependency strength (i.e., decreased conditional entropy or increased DI) has ​**a significant negative impact on DLLM performance, particularly when modeling long-range dependencies is required**​. This phenomenon is closely related to the inherent limitations of the parallel generation architecture, further validating our theoretical explanation of the PSC phenomenon. We have supplemented this experiment in Appendix J.

---

> > > ### Author Response · Authors · 2025-11-24
> > > **Response to Reviewer U4VP (Part3/3)**
> > >
> > > > **Question 4:** Data-side mitigation. Beyond prompting and decoding, could PSC be mitigated via dataset design? If synthetic datasets can be constructed, what would the ideal DLLM-friendly training data look like?
> > >
> > > **Answer 4: ​**Thank you for your insightful question. The essence of DLLM parallel generation is global optimization and the joint generation of all tokens. Sequential reasoning requires localized and conditional generation. PSC occurs precisely between these two paradigms. Therefore, we argue that an ideal, DLLM-friendly training dataset should not follow the traditional linear chain structure but should possess the characteristic of **"global conditional independence."** Our envisioned design for an ideal synthetic dataset is as follows:
> > >
> > > **​1.​Pre-planning Topology Structure**
> > >
> > > * ​**Data Format**​: `[Question] -> [Global Plan] -> [Detailed Steps] -> [Answer]`.
> > > * ​**Design Principle**​: The model is required to first output a high-level, low-dimensional "solution blueprint" or "global plan" in parallel before generating specific content. Once this plan is determined, it becomes the common condition for generating all subsequent detailed steps. Under this condition, the interdependency between individual steps is significantly reduced, making them suitable for parallel filling.
> > >
> > > **2. Modular Parallel Structure**
> > >
> > > * ​**Data Format**​: Explicitly label parallel blocks, e.g., `[Subtask A] || [Subtask B] || [Subtask C] -> [Result Aggregation]`.
> > > * ​**Design Principle**​: Decompose complex problems into independently solvable submodules and explicitly declare in the data that these submodules can be processed in parallel. This forcibly guides the model to learn the independence between tasks rather than an inherent sequential order.
> > >
> > > By systematically constructing synthetic data incorporating the above explicit independence assumptions and using it for fine-tuning the model, we can fundamentally reshape the model’s attention mechanism from the root. This enables the model to better adapt to the parallel reasoning paradigm of DLLMs, thereby fundamentally mitigating the PSC problem. We have added this content in Appendix F.8.

---

> > > > ### Comment · Reviewer_U4VP · 2025-11-27
> > > >
> > > > The authors’ response fully resolved my concerns, and I am willing to raise my score. Before doing so, however, I would like clarification regarding the novelty of PSC. Given that the related-work discussion is limited, I would like to know whether the authors are the first to propose PSC, or if prior works have discussed similar ideas. Since PSC is the core concept that supports the entire paper, understanding its originality is important.
> > > >
> > > > In addition, does this work imply that DLLMs are fundamentally incapable of performing self-correction?

---

> > > > > ### Author Response · Authors · 2025-11-27
> > > > > **Response to Reviewer U4VP**
> > > > >
> > > > > Thank you once again for your positive feedback and for raising these two profoundly important questions, which get to the very heart of our work. We are pleased to provide further clarification and hope to fully address your concerns.
> > > > >
> > > > >
> > > > > > 1. The Novelty of the Parallel-Sequential Contradiction (PSC)
> > > > >
> > > > > To the best of our knowledge, **this paper is indeed the first work that identifies, formalizes, and systematically investigates the "Parallel-Sequential Contradiction (PSC)" in DLLMs for Long Chain-of-Thought reasoning.**
> > > > >
> > > > > We acknowledge that prior research has touched upon various characteristics of DLLMs major in Efficiency and Effectiveness, for example:
> > > > >
> > > > > * **Efficiency Studies:** Works such as Mercury and Fast-DLLM [1] have primarily focused on the parallel decoding speed advantage of DLLMs.
> > > > > * **Reasoning Enhancement:** Works like d1 [2] and Trado [3] have sought to improve the reasoning performance of DLLMs through techniques like reinforcement learning or by leveraging sampling trajectories.
> > > > >
> > > > > However, these studies did not delve into the intrinsic mechanistic contradictions underlying their high performance. In contrast, our work is the first to reveal the fundamental tension: "the Parallel-Sequential Contradiction (PSC)". It is precisely this contradiction that dictates the behavioral patterns and capability ceilings of DLLMs in complex reasoning tasks.
> > > > >
> > > > > Therefore, the PSC serves as a novel theoretical lens. It​**​ provides a unified and fundamental explanation for understanding all behaviors of DLLMs in reasoning tasks**​, from the effectiveness of prompting strategies to the upper limits of scalability.
> > > > >
> > > > > > 2. Whether DLLMs Are Fundamentally Incapable of Self-Correction
> > > > >
> > > > > Our conclusion is **not that DLLMs are "fundamentally incapable" of self-correction.** Rather, we argue that ​**traditional, sequential self-correction paradigms are highly ineffective due to their conflict with the PSC**​. Hence, such findings outline **a practical route** to designing effective self-correction for DLLMs by circumventing or mitigating the PSC. In line with [4], effective self-correction can be elicited through ​**proofreading and revising the generated outputs**​, without relying on diffusion-based generation that exhibits strong PSC.
> > > > >
> > > > > In summary, our core thesis is this: Forcing DLLMs to correct on a fixed sequential order is ineffective due to their inherent PSC. However, ​**if we shift towards developing correction paradigms that synergize with their parallel architecture**​, **the potential for self-correction in DLLMs can be effectively realized. ​**Our work aims to pave the way for surpassing this very limitation by first clarifying it.
> > > > >
> > > > >
> > > > > We sincerely appreciate your prompting us to elaborate on these key points more clearly. We hope this response fully addresses your concerns and we look forward to your final assessment.
> > > > >
> > > > > [1]Wu C, Zhang H, Xue S, et al. Fast-dLLM: Training-free Acceleration of Diffusion LLM by Enabling KV Cache and Parallel Decoding[J]. arXiv preprint arXiv:2505.22618, 2025.
> > > > >
> > > > > [2] Zhao S, Gupta D, Zheng Q, et al. d1: Scaling Reasoning in Diffusion Large Language Models via Reinforcement Learning[J]. arXiv preprint arXiv:2504.12216, 2025.
> > > > >
> > > > > [3]Wang Y, Yang L, Li B, et al. Revolutionizing Reinforcement Learning Framework for Diffusion Large Language Models[J]. arXiv preprint arXiv:2509.06949, 2025.
> > > > >
> > > > > [4] Kim J, Kim S, Lee T, et al. Fine-Tuning Masked Diffusion for Provable Self-Correction[J]. arXiv preprint arXiv:2510.01384, 2025.

---

> > > > > > ### Comment · Reviewer_U4VP · 2025-11-27
> > > > > >
> > > > > > The authors’ response has fully addressed my concerns, and I now consider the paper both novel and solid. Accordingly, I have raised my score to 8.

---

### Official Review · Reviewer_dA1n · 2025-11-01

**Soundness:** 3
**Presentation:** 3
**Contribution:** 3
**Rating:** 4
**Confidence:** 2

**Summary:**

This paper studies the parallel-sequential tradeoffs in diffusion LLMs. That is, it observes that diffusion models inherently generate sequences in parallel. For tasks where there is a simple way to build up the sequence step by step (e.g., in reasoning), it is much harder to generate intermediate tokens without seeing the full context before the step. The paper call this "parallel-sequential contradiction".

The paper contains a broad empirical analysis to study properties of DLLMs from this perspective. First, it finds that compared to autoregressive LLMs, DLLMs often need more tokens and revert to sequential generation for more complex tasks. The paper explains this by saying that it is much easier to generate sequentially (lower entropy). Next, it looks at prompting strategies. Prompting strategies for reasoning are optimized for autoregressive models and hinder DLLMs; however, prompts that mention parallel reasoning appear to help.
Moreover, the paper shows that other common strategies that help CoT reasoning in ALLMs (autoregressive) do not help DLLMs. DLLMs seem to also in general make more mistakes.

The last part looks at scaling with respect to parallelism (more parallel samples), diffusion steps and sequence length. In particular more parallel samples lead to great improvements. More diffusion also helps up to a certain point, hence the paper suggests early stopping.

**Strengths:**

- The tension between the inherent parallelism and the sequential nature of certain reasoning tasks is a very relevant topic of study.

- The paper contributes a broad set of empirical results on the topic, which gives insights into trends and differences in behavior between autoregressive LLMs and DLLMs. The results are interesting.

- The results are structured clearly.

**Weaknesses:**

- The PSC, a core hypothesis of the paper, is defined a bit vaguely. Why is this a "contradiction"? I understand what the problem is -- parallel generation is disadvantageous if there are long-range dependencies in the sequence and there is a "simple" sequential generation rule. But it was vague to me at first reading, and I would use a different word than "contradiction" for this.
Similarly, what would then be "PSC-free", or "PSC-reduced"?

- The theory as is does not fully support the claims (and needs fixing), even though the claims are intuitive: The proof in Appendix B, if I understand it correctly, shows that for "sequential" tasks (input data distributions), the entropy of the k-th token given the first tokem (or step) is higher than for a "parallel" data distribution (which follows from the definitions of sequental and parallel via bounds on the probabilities). But it does not prove that a sequential *generation* would be better for either, as is claimed in Appendix B.5. For that, one would need to argue that the entropy increases with k, and that that makes it easier for the model to sequentially approximate p(S_k|S_1). In addition, the proof needs some polishing (see below).
While the theory needs to be fixed, the main contribution of the paper is empirical.

- Some of the observations are fairly straightforward (e.g., some of the behavior in scaling), as are the "fixes" of early stopping. The proposed prompt looks similar but simplified to the one right before. I think it is a great direction to think how to exploit both parallelism (in the correct way) and sequential reasoning, but how exactly the prompt is achieving this is not clear to me (even though it works).

- Some of the titles seem a bit unrelated to the text, at least in my understanding, e.g.:
* Sec. 4.1: if strategies for ALLMs do not work, does that really mean that DLLMs do not have sufficient capabilities for CoT? The only real limitation may be Fig. 6(a).
* Sec. 4.2.1: why "despite"? sampling multiple paths should not interfere with sequential vs. parallel reasoning.
* Sec. 4.2.2, 4.2.3: what does it mean for a scaling law to be "broken"? It is just an estimate how performance will scale with a resource. There is no diffusion scaling law for ALLMs, so what should it look like?
Moreover, the content of section 4.2.2 seems to be independent of PSC. Or what is the connection?

**Questions:**

- Is the function in Definition 1 (Appendix B) learnable, with epsilon going to zero? It may need at least an assumption on k to k+1 being continuous? Definition 1 even asks for the condition for $p(S_2|S_1)$.

- Appendix B: please check the notation, e.g.
* Lemma 1: the definition of the neighborhood (what is the radius? epsilon is now the cell size);
* eqn (8): this needs to be for some k (not any)
* Prop. 1: say what is $H_S$, $H_T$
* eqn 12: should the max be over k?

- Fig. 3b: why is there such a sharp drop in accuracy from 256 to 1024 tokens?

- parallel-encouraging prompting: Diff-MARP is less clear than MARP about what should be parallelized. Is MARP too complex? Or what do you mean by this: "This method balances parallel execution with simplicity, allowing for effective multi-step reasoning without overwhelming the model with overly complex tasks. Moreover, the model adjusts operation complexity dynamically." Why does Diff-MARP work better? This needs a bit more discussion in the main text, too.

- Figure 4,5 are a bit hard to grasp with the small amount of text in the main paper. What would you expect about e.g. token entropy from these strategies? compared to ALLMs? Is the x axis in the (c) figure percentage?

- Fig. 7 is not so clear to me. You could replace it by more explanatory text, as the text is very high-level.

- Fig. 9(e): it looks like in stage 3 the model has converged already.


Typos:
- Fig. 1: "surficial"
- takeaways page 5: redcued


I think this paper is interesting, and my score is due to the sloppiness in the theory part and the clarity in some places, as mentioned in my review. I am willing to change my score if these concerns are addressed.

---

> ### Author Response · Authors · 2025-11-24
> **Response to Reviewer dA1n (Part 1/5)**
>
> We sincerely thank you for your valuable comments and recognition of our work. As per your suggestions, we have responded to each point below.
>
> > **Question 1:** The PSC, a core hypothesis of the paper, is defined a bit vaguely. Why is this a "contradiction"?
>
> **Answer ​1:​** Thank you for your valuable feedback regarding the definition of PSC. We fully agree on the need to articulate this core concept more clearly. Below are some of our reflections, and we greatly look forward to hearing your further insights.
>
> Considerations regarding the term "Contradiction": We introduced the term "Parallel–Sequential Contradiction (PSC)" to capture a fundamental tension faced by diffusion models in reasoning tasks. The "contradiction" manifests at two levels:
>
> 1. **Mechanism level: ​**The inherent parallel decoding nature of diffusion models (predicting all tokens simultaneously) directly conflicts with the sequential dependency logic required for effective Chain-of-Thought (CoT) reasoning at the computational level.
> 2. **Behavioral level: ​**This conflict leads the model's generation process to oscillate between "following low-entropy sequential logic" and "falling into high-entropy parallel guessing," reflecting an inherent inconsistency.
>
> We use the term "contradiction" to emphasize this intrinsic, structural incompatibility. We have added relevant discussions in Section 2.3.  And we are also keen to know whether you believe alternative terms such as "conflict" or "tension" might more appropriately describe this phenomenon.
>
> > **Question 2:** What would then be "PSC-free", or "PSC-reduced"?
>
> **Answer 2:** Thanks for your insightful comments. We completely agree that this point could be phrased with greater precision. The core objective is not to entirely "eliminate" the parallel mechanism but to "mitigate" this contradiction through algorithmic design. Specifically, we aim to steer the model's parallel generation to better approximate sequential reasoning structures, thereby reducing the negative impact of PSC on performance.
>
> We hope these clarifications help convey the concept of PSC more accurately and look forward to your further guidance.
>
> > **Question 3:** If strategies for ALLMs do not work, does that really mean that DLLMs do not have sufficient capabilities for CoT?
>
> **Answer 3: ​**We sincerely thank the reviewer for their insightful comments. First, we would like to clarify that our prompting-based self-reflection and self-exploration methods can also, when appropriate, ​**conditionally encourage the model to generate additional reflections and explorations**​. This is consistent with Table 3: the constraint-guided reasoning prompting strategy we adopt does not activate any new PSCs and is well suited to DLLMs. In addition, we not only employ prompting-based methods but also use ​**autoregressive forcing strategies**​, where we append suffixes such as “wait...” to encourage the model to continue generating. This strategy likewise does not elicit any additional PSCs in DLLMs. Therefore, there is no issue that our findings depend on ineffective DLLM prompting.
>
> Moreover, we would like to emphasize that our core argument does not rely solely on the “ineffectiveness” of certain strategies. Instead, it is grounded in systematic diagnostics: through reflection, exploration, and sequential prompting (Section 3.3), we consistently observed **intrinsic limitations in DLLMs during Long CoT reasoning, such as high repetition rates and semantic rigidity, which are known to be central limitations of Long CoT reinforcement learning [1,2].** These limitations collectively explain the empirical findings in the paper, in particular the extremely shallow reasoning depth shown in Figure 6(a). Additionally, the CFRB/CIRB metrics we introduced (Figure 6) quantitatively confirm that DLLMs operate within a narrower viable reasoning range. Thus, the ineffectiveness of strategies is the observed phenomenon, the shallow reasoning depth is the outcome, and the PSC mechanism deficiency is the underlying cause. Together, these three aspects form a complete chain of evidence for the fundamental challenges DLLMs face in Long CoT reasoning.
>
> [1] Chen Q, Qin L, Liu J, et al. Towards reasoning era: A survey of long chain-of-thought for reasoning large language models[J]. arXiv preprint arXiv:2503.09567, 2025.
>
> [2] Xu F, Hao Q, Zong Z, et al. Towards large reasoning models: A survey of reinforced reasoning with large language models[J]. arXiv preprint arXiv:2501.09686, 2025.

---

> ### Author Response · Authors · 2025-11-24
> **Response to Reviewer dA1n (Part 2/5)**
>
> > **Question 4:** Why "despite" in Section 4.2.1? Sampling multiple paths should not interfere with sequential vs. parallel reasoning.
>
> **Answer 4: ​**Thanks for your detailed suggestion. We sincerely believe that there are some misunderstandings. Our core research question here is whether DLLMs can still adhere to the "parallel scaling law" under the inherent PSC constraint. To express this more clearly, we will refine the wording in the revised version of the paper to:"​**PARALLEL SCALING LAW HOLDS UNDER PSC CONSTRAINT**​"
>
> > **Question 5:** What does it mean for a scaling law to be "broken" in Sections 4.2.2 and 4.2.3? It is just an estimate how performance will scale with a resource. There is no diffusion scaling law for ALLMs, so what should it look like?
>
> **Answer 5: ​**Thank you for your valuable suggestion. Our intention is not to challenge the scaling law itself, but to highlight that in DLLMs, the scaling behavior deviates from the typical pattern: it is ​**constrained by PSC and exhibits a saturation upper bound rather than unbounded growth**​. Accordingly, we have replaced the term “BROKEN” with more precise wording and revised the relevant section titles in the updated manuscript.
>
> * New Title (4.2.2): "DIFFUSION SCALING IS CONSTRAINED BY PSC"
> * New Title (4.2.3): "SEQUENTIAL SCALING IS EQUALLY CONSTRAINED BY PSC"
>
> > **Question 6:** The content of Section 4.2.2 seems to be independent of PSC. Or what is the connection?
>
> **Answer 6:** Thank you for your thoughtful response. The diffusion scaling law originally posits that model performance should continuously improve with an increase in diffusion steps. However, we observe that once the number of diffusion steps exceeds a certain threshold, performance gains tend to saturate or even decline. This phenomenon is precisely attributed to the PSC: excessive parallel refinement steps interfere with the sequential reasoning chain inherently relied upon by the model, thereby hindering the sustained effectiveness of scaling.
>
> Specifically, when the number of diffusion steps becomes excessively high (e.g., increasing from 512 to 1024 steps), the model’s accuracy drops from 44.92% to 44.43%. This "over-diffusion" phenomenon fundamentally stems from the inability of the parallel generation mechanism to effectively coordinate long-range reasoning dependencies, which can be regarded as a manifestation of PSC in the scaling process.
>
> In summary, we aim to demonstrate that **PSC is not only a challenge at the reasoning level but also constrains the scaling behavior of diffusion models at a macroscopic level.**
>
>
> > **Question 7:** Why is there such a sharp drop in accuracy from 256 to 1024 tokens?
>
> **Answer 7:** Thank you for your detailed feedback. We observe that the average solution length in the BigGSM dataset falls within the range of 256–512 tokens. This implies that:
>
> * For sequences shorter than 256 tokens, the effectiveness of diffusion improves as length increases. This is expected, as the model approaches its optimal reasoning length. As the sequence length nears the dataset's "sufficient length" threshold (approximately 256 tokens here), accuracy improves
>
> However, beyond 512 tokens, most of the additional tokens are empty or redundant and require repeated remasking for validation. This leads to the following issues:
>
> * **Iteration Dilution and Over-Diffusion:** When the sequence length exceeds what the task requires, an increasing number of blank or redundant positions are incorporated into the diffusion process. Denoising iterations become distributed over a longer sequence, making it more difficult to maintain global coherence, and performance begins to decline.
> * **Uncertainty in Termination Signal: ​**When the length significantly exceeds what is needed, the model tends to "fill the space" rather than reliably generate an EOS (end-of-sequence) token. This introduces redundancy and interference, ultimately reducing final judgment accuracy. (The larger the ratio of `max token length / diffusion steps`, the longer the generated sequence tends to be. When this ratio exceeds 16, the model rarely stops to produce a conclusion.)

---

> > ### Author Response · Authors · 2025-11-24
> > **Response to Reviewer dA1n (Part 3/5)**
> >
> > > **Question 8:** Why does Diff-MARP work better?
> >
> > **Answer 8:  ​**Thank you for your valuable comments. The performance improvement of Diff-MARP hinges critically on the prompt **“Reasoning in parallel.”** This instruction directly engages the model’s capacity for parallel reasoning. Diff-MARP provides a structured implementation of this idea, imposing an upper bound on the number of operations to avoid parallel overload and thereby achieving more robust performance than the base instruction alone.
> >
> > To further validate this, we replace “Reasoning in parallel.” with “Reasoning step-by-step sequentially.” as Seq-MARP. We observe a substantial performance decrease under this replacement. Therefore, we conclude that**​ the parallel reasoning instruction** is the primary driver of the observed performance gain.
> >
> > | Prompt | BigGSM Acc. |
> > | ------------------ | ----------------------- |
> > | Seq-MARP         | 41.91                 |
> > | Diff-MARP        | 47.21                 |
> > ||
> >
> > Table 2: The performance comparison between Seq-MARP and Diff-MARP.
> >
> > > **Question 9:** Figure 4,5 are a bit hard to grasp with the small amount of text in the main paper. What would you expect about e.g. token entropy from these strategies? compared to ALLMs? Is the x axis in the (c) figure percentage?
> >
> > **Answer 9: ​**Thank you for your constructive suggestions. Regarding the token entropy of these strategies, an effective reflection strategy is generally expected to ​**drive the model toward lower entropy and higher certainty**​, thereby yielding​**​ more confident outputs from DLLMs through self-reflection**​. However, current DLLMs cause increased entropy, which suggests that current reflection mechanisms in DLLMs provide only limited, surface-level optimization and remain restricted to a relatively narrow semantic scope. We have added relevant notes in Section 4.1.
> >
> > Moreover, in Figure (c), the x-axis denotes the proportion of trajectories categorized as new paths or error paths. We instruct human annotators to determine, for each reflection or exploration trajectory, whether it is repetitive with respect to the preceding (baseline) trajectory (True/False) and whether its final outcome is correct (True/False). We then compute the proportion of True labels, which is reported on the x-axis as percentages. We have added this description to the caption of Figure 4.
> >
> > > **Question 10:** Fig. 7 is not so clear to me. You could replace it by more explanatory text, as the text is very high-level.
> >
> > **Answer 10: ​**Thanks for your detailed feedback. Figure 7 illustrates the three primary scaling directions for DLLMs proposed in our work:
> >
> > * **Diffusion Scaling:** Increasing the number of denoising reasoning steps in the model. The stacked MASK blocks represent multiple, deeper iterations of denoising within a single generation process.
> > * **Parallel Scaling: ​**Generating multiple distinct output candidates simultaneously in parallel. The adjacent output blocks at the top represent the concurrent production of multiple output variants.
> > * **Sequential Scaling: ​**The visualization is designed to emphasize the step-by-step output generation process. The arrow direction explicitly illustrates this incremental reasoning progression.
> >
> > We have added a detailed description in Section 4.2 of the paper.
> >
> > > **Question 11:** Fig. 9(e): it looks like in stage 3 the model has converged already.
> >
> > **Answer 11:** Thanks for your thoughtful comment. Yes, the paper intends to convey this idea. We identify three stages in the diffusion process of DLLMs: ​(1) Sequential Scaling, (2) Compression, and (3) Convergence​. In the first stage, increasing the number of diffusion steps yields steady performance gains but also lengthens the solutions. In the second stage, the model compresses its reasoning by removing redundant steps, thereby improving efficiency. In the third stage, the model converges to a stable strategy that preserves high performance while reducing computational cost.
> >
> > We have emphasized in Section 4.2.3 of the paper.
> >
> > > **Question 12:** Typos:Fig. 1: "surficial"，takeaways page 5: redcued
> >
> > **Answer 12:** Thank you for pointing this out. We have made the corresponding corrections in the revised version of the paper.

---

> > > ### Author Response · Authors · 2025-11-24
> > > **Response to Reviewer dA1n (Part 4/5)**
> > >
> > > > **Question 13:** It is a great way to think about how to exploit both parallelism (in the correct way) and sequential reasoning, but how exactly the prompt is achieving this is not clear to me (even though it works).
> > >
> > > **Answer 13:** Regarding how to utilize parallelism and sequential reasoning in prompting, we provide a case study of stepwise reasoning using the Diff-MARP strategy.
> > >
> > > Using the Diff-MARP Strategy, we can obtain the solution that:
> > >
> > > ```
> > > Step 1: Calculate the total sleep time from 10 p.m. to 6 a.m.
> > > Kim's usual sleep time is from 10 p.m. to 6 a.m., which is 8 hours. Since there are 60 minutes in an hour, her usual sleep time is 8 * 60 = 480 minutes.
> > > Step 2: Calculate the time she spent sleepwalking.
> > > Kim was sleepwalking from 2:15 to 2:38 am, which is 23 minutes.
> > > ...
> > > ```
> > >
> > > We observe that each computational step is explicitly annotated and explained. This decomposition improves transparency in the reasoning process, making the logic from start to finish easy to follow. **DLLMs can perform parallel reasoning within each step while maintaining only limited sequential dependencies between steps.**
> > >
> > > Without the Diff-MARP Strategy, we can obtain the solution that:
> > >
> > > ```
> > > Kim usually goes to sleep at 10 p.m. and wakes up at 6 a.m. This is a total of 8 hours.
> > > On the day in question, she woke up 5 minutes earlier than usual to go to the bathroom.
> > > She was sleepwalking from 2:15 to 2:38 am, which is a total of 23 minutes.
> > > ...
> > > ```
> > >
> > > When the Diff-MARP strategy is not used, the numerical results remain identical, but the procedure is more compact and the reasoning less systematically organized, so the parallel–sequential pattern is left random and can even collapse into purely sequential generation. The key distinction is that Diff-MARP explicitly structures the intermediate sub-steps, thereby yielding a clearer and more automatic balance between parallel and sequential processing.
> > >
> > > In the revised version of the paper, we have further clarified how this strategy enhances the model’s overall efficiency and have made these points more explicit.The detailed description of this part has been added in Appendix F.4 of the paper.
> > >
> > > > **Question 14:** Questions about the proof in Appendix B.
> > >
> > > **Answer 14:** We are grateful for your rigorous examination of our proof. Your interpretation is correct: our previous formulation only established the hardness of parallel prediction, but did not formally connect this to the advantage of sequential generation, and our original learnability definition implicitly allowed for chaotic, unlearnable cases. We have therefore revised Appendix B in two ways.
> > >
> > > 1. **Completing the Proof (Why Sequential is Better):** To support the claim in B.5, we explicitly introduce the **Chain Rule of Entropy**. We posit that while the long-range conditional entropy $H(S_k|S_1)$ is high (as proved), the divided sub-step-wise conditional entropy $H(S_{t+1}|S_{\le t})$ is low. **If each step is relatively certain** (with low conditional entropy)​, **then learning through many small, low-entropy problems** ​(autoregressive)​ **is much more efficient than solving a large, high-entropy problem all at once** (directly jumping from $S_1$ to $S_k$)
> > > 2. **Refining Definition 1 (Learnability):** We agree that strictly satisfying Eq. (3) for *any* perturbation implies a chaotic, unlearnable system. We **relax Definition 1** to assume **Local Continuity**: specifically, that the transition **$S_t \to S_{t+1}$** is continuous and learnable (stable against small \$\\epsilon\$), whereas the compounded effect over $k$ steps leads to the high divergence described. This ensures the task is solvable step-by-step but intractable for skip-step parallel prediction.

---

> > > > ### Author Response · Authors · 2025-11-24
> > > > **Response to Reviewer dA1n (Part 5/5)**
> > > >
> > > > > **Question 15:** Check the notation, e.g.
> > > > >
> > > > > * Lemma 1: the definition of the neighborhood (what is the radius? epsilon is now the cell size);
> > > > > * Eq 8: this needs to be for some k (not any)
> > > > > * Prop. 1: say what is HS, HT
> > > > > * Eq 12: should the max be over k?
> > > >
> > > > **Answer 15:** We are grateful for your meticulous review of our mathematical proofs. You correctly identified several notational inconsistencies and logical leaps. We will revise Appendix B as follows:
> > > >
> > > > 1. **Learnability & Definition 1:** We will relax Definition 1 to assume that the step-wise transition **$P(S_{t+1}|S_t)$** is **continuous and learnable** (low entropy), while the long-range mapping **$S_1 \to S_k$** exhibits high sensitivity (high entropy). This ensures the task is solvable via autoregression, supporting our conclusion without implying unlearnable chaos.
> > > > 2. **Notation Separation (Lemma 1):** We will strictly distinguish between the perturbation radius （Denoted by  ε） and the discretization mesh size (Denoted by  δ）
> > > > 3. **Quantifiers (Eq 8 & 12):**
> > > >    - For Eq (8), we will correct the quantifier  to**​ "for some ​k" ​**​(indicating stable intermediate/final states in parallel tasks).
> > > >    - For Eq (12), we will clarify that the maximization is performed over the neighborhood of $s_1$ for **a fixed target step ​k.**
> > > > 4. **Undefined Symbols:** We will explicitly define $H_S$ and $H_T$ as the conditional entropies under the Serial and Parallel data distributions, respectively.
> > > >
> > > > We believe these rigorous corrections will make the mathematical proof sound and fully supportive of our empirical findings.
> > > > We have revised the aforementioned sections in Appendix B.
> > > >
> > > > Hope our clarifications can address your concerns, and we **have followed your suggestions** to enrich our work point-by-point in the revised version. **We sincerely hope that you can reconsider our work in light of these clarifications**.

---

### Official Review · Reviewer_pC3J · 2025-11-04

**Soundness:** 3
**Presentation:** 3
**Contribution:** 3
**Rating:** 8
**Confidence:** 4

**Summary:**

This paper investigates the fundamental reasoning limits of DLLMs. It identifies a core Parallel–Sequential Contradiction (PSC), where the parallel decoding nature of DLLMs conflicts with the inherently sequential structure of reasoning. Through behavioral and scaling analyses, the authors show that DLLMs revert to autoregressive-like behavior in complex reasoning tasks and face efficiency losses. They propose several mitigation methods, which alleviate PSC effects and improve reasoning accuracy by up to 5% across benchmarks such as BigGSM and GSM8K.

**Strengths:**

* The identification and formalization of the PSC provide a theoretically grounded explanation for DLLMs’ inefficiencies in sequential reasoning, which offers an important new perspective for future research.
* The authors decompose DLLM inference into three scaling dimensions and systematically examine their performance and upper bounds. As far as I know, this is the first work to systematically study scaling on text diffusion models.
* The empirical analysis is comprehensive and the figure is well illustrated. The motivation and the presentation are both good.

**Weaknesses:**

* The authors claim that they use the default decoding settings for all models. However, as claimed in [1], the decoding method is vital in diffusion model generation. The authors could try more decoding method (entropy-based, or top-p margin based in [1]), which can help determine whether PSC is an inherent property of DLLMs’ structure or merely a byproduct of the default decoding method.
* Missing citations and discussions. [2] also discusses the scaling behavior of diffusion models, which shares some same views with the paper.

[1] Train for the Worst, Plan for the Best: Understanding Token Ordering in Masked Diffusions. Arxiv 2502.06768
[2] Inference-Time Scaling for Diffusion Models beyond Scaling Denoising Steps. Arxiv 2501.09732

**Questions:**

* How does PSC interact with post-training method? Does fine-tuning or RL training reduce or exacerbate the contradiction?

---

> ### Author Response · Authors · 2025-11-24
> **Response to Reviewer pC3J**
>
> We sincerely thank you for your valuable comments and recognition of our work. As per your suggestions, we have responded to each point below.
>
> > **Question 1**: As claimed in [1], Is PSC an inherent property of DLLMs’ structure or merely a byproduct of the default decoding method?
>
> **Answer 1**: Thanks for your insightful feedback. Following your suggestion, we conducted prompt experiments on BigGSM using two decoding methods, entropy and topk_margin. As shown in Table 1 below, the results are **consistent with the experimental results** in the paper, which demonstrate that PSC is an inherent property of the DLLM architecture rather than a byproduct of the default decoding method. We have added this section to Appendix F.7.
>
>
>
> | Accuracy       | Entropy-based decoding | topk\_margin decoding |
> | ---------------- | ------------------------ | ----------------------- |
> | baseline       | 41.15                  | 41.97                 |
> | zero-cot       | 41.15                  | 39.84                 |
> | Least to most  | 34.75                  | 32.62                 |
> | Plan and solve | 35.25                  | 41.48                 |
> | complex-cot    | 41.80                  | 43.44                 |
> | MARP           | 42.95                  | 46.07                 |
> | DIFF-MARP      | 47.21                  | 48.19                 |
> ||
>
> Table 1: Accuracy of different prompting and decoding methods on BigGSM based on Dream.
>
> [1] Train for the Worst, Plan for the Best: Understanding Token Ordering in Masked Diffusions. Arxiv 2502.06768
>
>
>
> > **Question 2:** Missing citations and discussions. [2] also discusses the scaling behavior of diffusion models, which shares some same views with the paper.
>
> **Answer 2:** Thanks for your valuable feedback.We have carefully studied this work [2] and fully agree with their perspective, as it bears significant relevance to our discussion on the "scaling behavior of diffusion models." We discuss this in Section 4.2.3. However, we sincerely believe that previous work has focused exclusively on ​**diffusion scaling in image generation with pure diffusion models**​, whereas ​**we are the first to investigate diffusion scaling in diffusion large language models**​.
>
> [2] Inference-Time Scaling for Diffusion Models beyond Scaling Denoising Steps. Arxiv 2501.09732
>
> > **Question 3:** How does PSC interact with post-training methods? Does fine-tuning or RL training reduce or exacerbate the contradiction?
>
> **Answer 3:** Thank you for your thoughtful suggestions. To investigate the impact of post-training methods on the PSC phenomenon, we conducted a comparative study between the LLaDA-8B-Instruct model and its reinforcement learning-finetuned version (d1), using consistent experimental parameters.
>
> We sampled 100 examples from the BigGSM dataset and recorded variations in the decoding order during the generation process. Specific evaluation metrics include:
>
> * **Inversion Pair Ratio: ​**This refers to cases where "a token in an earlier position is generated later than a token in a later position." If the position ​i<​j​, but the generation step step[​i]>step[​j], it is considered an inversion pair. A higher ratio indicates more severe decoding inversion
> * **Spearman’s Rank Correlation Coefficient: ​**This is used to measure the consistency between the model’s generation order and the natural language order (left-to-right). A higher coefficient indicates that the model tends to follow the natural order more closely.
>
> || Inversion Pair Ratio | Spearman’s Cor |
> | ------------------- | ---------------------- | ----------------- |
> | LLaDA-8B-Instruct | 3.41%                | 98.62%          |
> | d1                | 6.65%                | 97.27%          |
> ||
>
> Table 2: The PSC analysis based on LLaDA-8B-Instruct and its RL version (d1).
>
> The results indicate that after reinforcement learning training, the model's inversion pair ratio increased from 3.41% to 6.65% (more parallel), while the Spearman coefficient decreased from 98.62% to 97.27% (reduced fluency). This suggests that reinforcement learning training has intensified sequential conflicts during decoding, making the PSC phenomenon more pronounced. We have included this experiment in Appendix I.

---

### Author Response · Authors · 2025-11-24
**General Response**

We sincerely thank all reviewers for their valuable and thoughtful feedback.

1. We are encouraged that the reviewers consistently **recognized the identification and formalization of the Parallel-Sequential Contradiction (PSC) as a novel and important perspective**, providing a theoretically grounded explanation for the inefficiencies of DLLMs in sequential reasoning (Reviewer #pC3J, #iUzq, #eVwU), and **acknowledged its value in inspiring future research** (Reviewer #pC3J, #eVwU).
2. We are also pleased that the reviewers **highly valued the comprehensive empirical analysis** (Reviewer #pC3J, #dA1n, #U4VP, #eVwU), **finding the experimental design complete and well-executed** (Reviewer #U4VP), the results clear and interesting (Reviewer #dA1n, #eVwU), and the systematic exploration of the scaling dimensions of DLLMs noteworthy (Reviewer #pC3J).
3. Furthermore, the reviewers **acknowledged the clear motivation and presentation of the paper** (Reviewer #pC3J, #dA1n), as well as the convincing nature of the mathematical derivations (Reviewer #iUzq).

We will address all concerns to polish our work according to reviewers’ comments in the next version.

---

### Note · Program_Chairs · 2026-01-17
**Submission Desk Rejected by Program Chairs**

The following references in this submission do not refer to real documents and/or have major errors in bibliographic information:

 Yatao Nie, Jie Chen, Yufan Zhang, et al. Large language diffusion with masking. arXiv preprint, 2025b. URL https://arxiv.org/abs/2502.09992.